# Conjugated cross-linked phosphine as broadband light or sunlight-driven photocatalyst for large-scale atom transfer radical polymerization

Wei-Wei Fang[1,4], Gui-Yu Yang[1,4], Zi-Hui Fan[1], Zi-Chao Chen[1], Xun-Liang Hu[2], Zhen Zhan[2], Irshad Hussain [3], Yang Lu [1], Tao He[1] ✉ & Bi-En Tan [2] ✉

The use of light to regulate photocatalyzed reversible deactivation radical polymerization (RDRP) under mild conditions, especially driven by broadband light or sunlight directly, is highly desired. But the development of a suitable photocatalyzed polymerization system for large-scale production of polymers, especially block copolymers, has remained a big challenge. Herein, we report the development of a phosphine-based conjugated hypercrosslinked polymer ($PPh_3$-CHCP) photocatalyst for an efficient large-scale photoinduced copper-catalyzed atom transfer radical polymerization (Cu-ATRP). Monomers including acrylates and methyl acrylates can achieve near-quantitative conversions under a wide range (450–940 nm) of radiations or sunlight directly. The photocatalyst could be easily recycled and reused. The sunlight-driven Cu-ATRP allowed the synthesis of homopolymers at 200 mL from various monomers, and monomer conversions approached 99% in clouds intermittency with good control over polydispersity. In addition, block copolymers at 400 mL scale can also be obtained, which demonstrates its great potential for industrial applications.

Sunlight is a typical abundant and sustainable energy source, and developing solar-driven reactions has always been a frontier research area from a green process perspective[1]. Solar energy has been harvested to drive many chemical reactions[2] including $CO_2$ transformation[3,4], $CH_4$ activation[5–9], $NH_3$ synthesis[10,11], water splitting[12–15], and organic molecules synthesis, etc[16,17]. However, most large-scale photocatalytic reactions are hampered till present due to the difficulty in developing efficient photocatalysts, and there are only a few reports on the solar hydrogen production[18,19]. Therefore, there is a great impetus to develop large-scale production using sunlight in many chemical processes, because they are more sustainable, energy-saving, and environmental-friendly.

Industry has a strong interest in radical polymerization (RP) due to its typical advantages, such as a wide choice of monomers, easily available catalysts, and mild reaction conditions, etc[20]. RP is one of the most important polymerization techniques, which produced ~45% (or 170 million metric tons) of plastics in 2015[21,22]. However, the irreversible chain termination and transfer reactions occurred during RP,

[1]School of Chemistry and Chemical Engineering, Anhui Province Key Laboratory of Advanced Catalytic Materials and Reaction Engineering, Hefei University of Technology, Hefei, Anhui 230009, PR China. [2]School of Chemistry and Chemical Engineering, Key Laboratory of Material Chemistry for Energy Conversion and Storage Ministry of Education, Hubei Key Laboratory of Material Chemistry and Service Failure, Huazhong University of Science and Technology, Wuhan, Hubei 430074, PR China. [3]Department of Chemistry & Chemical Engineering, SBA School of Science & Engineering, Lahore University of Management Sciences (LUMS), Lahore Cantt, Lahore 54792, Pakistan. [4]These authors contributed equally: Wei-Wei Fang, Gui-Yu Yang. ✉e-mail: taohe@hfut.edu.cn; bien.tan@mail.hust.edu.cn

which made it less competitive to meet the increasing demand for advanced functional polymeric materials, such as polymers with pre-determined molecular weight, narrow distribution, pre-designed block sequence, and complex architecture, etc.[23]. The reversible deactivation radical polymerization (RDRP) being developed over the past 25 years, could combine the controlled polymerization nature and precision polymer functionalization by strictly controlling the dynamics of dormant-active equilibria and reducing the irreversible termination between active propagating radicals. RDRP is now a part of the industrial chemist's toolbox for precision polymers and block copolymers synthesis[24]. Products (mainly block copolymers) fabricated through RDRP (e.g., pigment dispersants, adhesives, sealants, surface modifiers, and chromatographic supports, etc.) already played key roles in many areas including coating, oil field, construction, printing industry, energy, automotive, aerospace, personal care, and biomedical markets, etc.[25].

Great efforts have been made to explore photo-mediated/induced RDRP to develop green and sustainable processes during the past decade[26]. Both the photocatalyzed atom transfer radical polymerization (ATRP) and photo-induced electron/energy transfer-reversible addition-fragmentation chain transfer polymerization (PET-RAFT) offered novel routes to synthesize various polymers, demonstrating typical advantages over the conventional thermal process. In this regard, researchers have endeavored to develop cost-effective photocatalysts with low toxicity, high activity, and the ability to absorb visible light at longer wavelengths[27]. Meanwhile, the efforts development of materials/methodology for large scale synthesis with a good control over the polymer's molecular weight and dispersity have always been on rise. As reported, several photo RDRP systems have shown marvelous prospects of scaling up reactions with the rapid development of microfluidic engineering[28–30]. Related products such as PAO-b-PPEGMA showed great potential application in boosting uranium harvesting from seawater[29]. From the industrial application point of view, large-scale production of block copolymers is clearly an area worth continuous exploring.

There are several typical barriers in RDRP using sunlight directly. For example, in practical applications, different from laboratory light sources, the illumination intensity of sunlight is much higher[31]. This can lead to the saturation of most organic dyes, resulting in the irreversible deterioration of the excited molecules and their photobleaching[32,33]. Meanwhile, the high-energy photons in the solar spectrum can also cause side reactions such as self-initiation of monomers and degradation of polymers containing unstable groups in main chains. These features greatly limit the application of photomediated RDRP (mainly PET-RAFT) using sunlight, considering most of the photocatalysts being applied are dyes or related compounds[34–38].

In organocatalyzed ATRP, polymerization being performed under sunlight is less efficient than that under white light[39,40]. Early choice of inorganic semiconductors such as ZnO and $TiO_2$ nanoparticles were not compatible with sunlight-driven photopolymerization due to the broad energy band (UV region) and low solar utilization efficiency (~5%)[41]. An alternative could be carbon-based materials (carbon dots (CDs), or conjugated microporous polymers (CMPs), etc.), which possess fascinating photophysical properties and photostability, and therefore could be used as photocatalysts in photo-mediated RDRP[42–45]. Benefitting from heterogeneous nature, photocatalysts such as CMPs of phenothiazine (PTZ-CMP) can not only enable easy separation and efficient reusability in Cu-ATRP, but also offer polymers with high conversion and well-controlled molecular weight under green light irradiation[44]. However, it has been very challenging to achieve ideal monomer conversions (99%) within a single solar illumination period (6–8 h) using carbon-based materials as photocatalyst[45]. Metal organic frameworks (MOFs) with various functional building blocks and dimension such as porphyrinic zirconium MOF (MOF-525(Zn)) and two-dimensional (2D) ZnTCPP nanosheets

(TCPP: 5,10,15,20-(tetra-4-carboxyphenyl) porphyrin), have been employed as highly efficient heterogenous photocatalysts in stereolithographic 3D printing, and created 3D polymeric objects using visible light irradiation in short period[46,47]. However, sunlight was not applied in these polymerizations. In other systems, although the localized photothermal effect of photocatalysts ($Ag_3PO_4$, N-doped CDs, etc.) facilitated nearly quantitative monomer conversions under sunlight[48], they were subjected to many disadvantages such as the tedious separation process, narrow choice of monomers (only highly reactive monomers can be used) and relatively broad dispersity of the synthesized polymers[49,50].

In this regard, we report the synthesis of phosphine-based conjugated hyper crosslinked polymer ($PPh_3$-CHCP) photocatalyst and its application for the first persistent large-scale sunlight-driven Cu-ATRP with limited $O_2$ tolerance (without deoxygenation procedure) (Fig. 1). The raw materials and the reaction conditions for the synthesis of photocatalyst are feasible to be upscaled at an industrial level. Most polymerizations were completed at 99% monomer conversions with a good control, which were conducted over a wide range of 450–940 nm light irradiation due to the inherent broad light absorption of $PPh_3$-CHCP, and the light intensities applied were very low. The heterogeneous nature of the photocatalyst allowed easy separation and reuse in multiple cycles with the retention of high photocatalytic efficiency. In sunlight-driven polymerizations, monomers (e.g., methyl acrylate (MA) and methyl methacrylate (MMA)) could achieve near quantitative conversion in a single solar illumination period (6 h with cloud intermittency) at scaled-up production of up to 200 mL (the largest reaction scale being reported as of now), with good control over dispersity. The block copolymer of PMA-b-PMMA could be in situ synthesized at 400 mL scale using the pre-prepared PMA macroinitiator under blue light irradiation (also the largest scale being reported until present). The development of this photocatalyst and the related protocol is very likely to positively impact photocatalyzed radical polymerization process, together with the potential industrial application of green and sustainable RP process.

## Results

### Synthesis and characterization of the photocatalyst

$PPh_3$-HCP was synthesized using Friedel-Crafts alkylation reaction between triphenylphosphine (TPP) and dimethoxybenzene (DMB) in the presence of iron (III) chloride. The $PPh_3$-HCP were subsequently treated with $NaBH_4$ to produce $PPh_3$-CHCP (Fig. 1a and Supplementary Figs. 1 and 2). Compared with $PPh_3$-HCP, the emerging peak of $PPh_3$-CHCP near 33 ppm indicated the interactions among phosphorous atom and introduced groups during treatment (solid-state $^{31}P$ nuclear magnetic resonance (NMR) spectrum in Fig. 2a). Thermogravimetric (TG) results suggested the presence of $PPh_3$-CHCP residue even at 800 °C (Supplementary Fig. 3). X-ray photoelectron spectroscopy (XPS) spectroscopy in Fig. 2c, d revealed that the oxidation states of C, P, Cl, and O elements in both $PPh_3$-HCP and $PPh_3$-CHCP were identical. The binding energy of P in $PPh_3$-CHCP was found to be 133.1 eV, indicating the formation of P=O, which may be resulted from the oxidization reaction between ferric chloride and triphenylphosphine in the presence of dissolved oxygen in the reaction media[51]. Compared with $PPh_3$-HCP, the peaks in $PPh_3$-CHCP at around 497.0 eV and 1072 eV could be assigned to Na. Considering there was no oxidation state change of phosphorus element, electrostatic interaction of $Na^+$ with phosphorous may result in the intercalation of $Na^+$ into electron-rich $PPh_3$-CHCP framework. Energy dispersive spectroscopy (EDS) analysis further indicated the existence of Na (green) in the frameworks (Supplementary Fig. 4).

$CO_2$ sorption isotherms revealed that $PPh_3$-CHCP possessed a Brunauer-Emmett-Teller (BET) surface area of 119 $m^2/g$ (Supplementary Fig. 5). The $CO_2$ adsorption-desorption isotherm was a typical II-type curve, which showed a hysteresis loop at low relative pressures

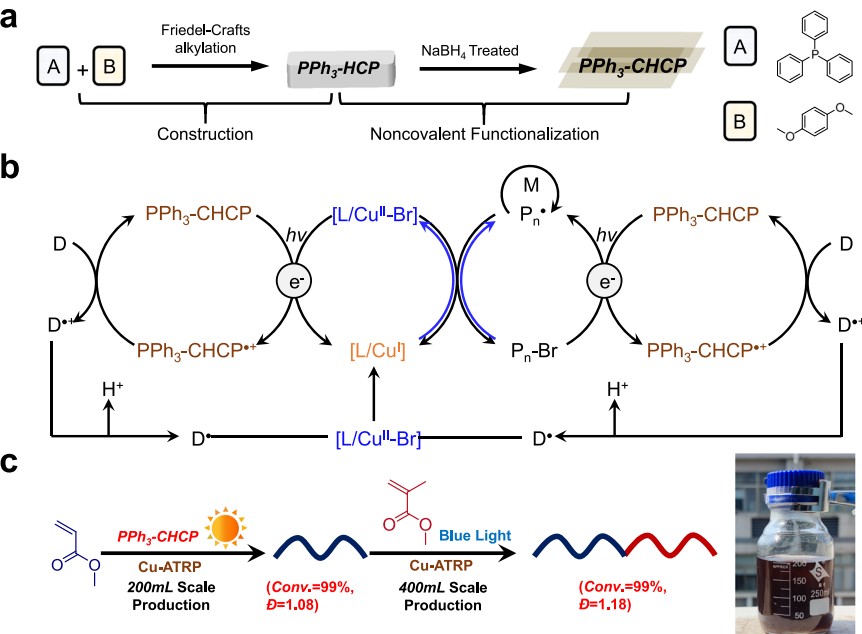

**Fig. 1 | Development of photocatalyst for large scale sunlight-driven Cu-ATRP.** **a** Strategy for the fabrication of PPh₃-CHCP photocatalyst. **b** Generation of ATRP activators *via* photoredox reactions. **c** Photoinduced Cu-ATRP in the presence of PPh₃-CHCP (inserted photo: 200 mL reaction scale of PMA under sunlight irradiation).

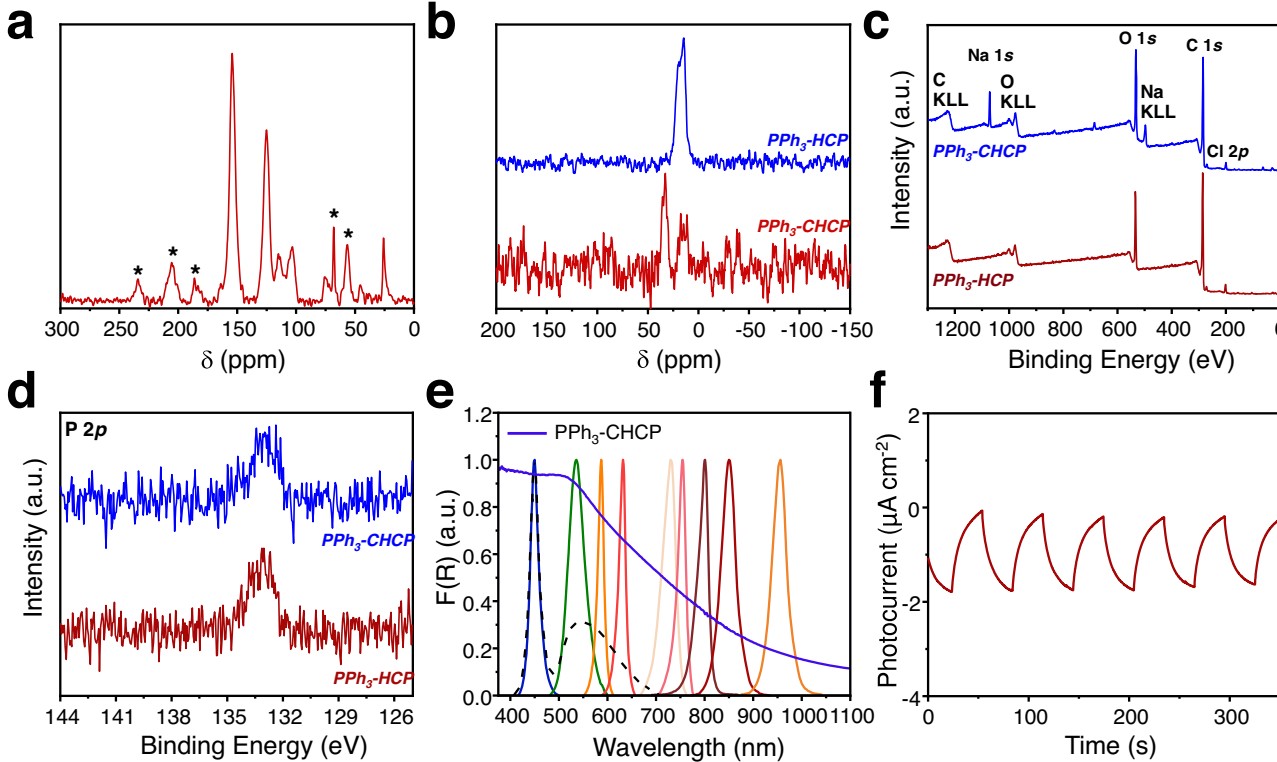

**Fig. 2 | The structure characterization and photoelectric performance of the PPh₃-CHCP.** **a** Solid-state $^{13}$C NMR spectrum of PPh₃-HCP. Asterisks denote spinning sidebands. **b** Solid-state $^{31}$P NMR spectra of PPh₃-HCP and PPh₃-CHCP respectively. **c**, **d** Relative XPS survey spectra of PPh₃-HCP and PPh₃-CHCP respectively. **e** UV-Vis-NIR diffuse reflectance spectrum of the photocatalyst overlaid with the emission spectra of the light sources, including blue ($\lambda_{max} = 455$ nm), green ($\lambda_{max} = 540$ nm), orange ($\lambda_{max} = 590$ nm), red ($\lambda_{max} = 630$ nm), 730 nm, 760 nm, 800 nm, 850 nm, 940 nm, and white light respectively. **f** Photocurrent response curve of PPh₃-CHCP.

**Table 1 | Results of ATRP of MA using PPh₃-CHCP photocatalyst in the presence of different ligands under green light irradiation**

| Entry | Ligand | CuBr₂/L | PPh₃-CHCP (mg/mL) | Solvent | Time (h) | Conv (%) | $M_{n,th}$ | $M_n$ | Đ |
|---|---|---|---|---|---|---|---|---|---|
| 1 | Me₆TREN | 1/1 | 0.5 | DMSO | 8 | 0 | – | – | – |
| 2 | Me₆TREN | 1/2 | 0.5 | DMSO | 8 | 0 | – | – | – |
| 3 | Me₆TREN | 1/3 | 0.5 | DMSO | 8 | 99 | 17,300 | 17,600 | 1.06 |
| 4 | Me₆TREN | 1/4 | 0.5 | DMSO | 8 | 99 | 17,300 | 18,700 | 1.06 |
| 5 | Me₆TREN | 1/3 | 0.25 | DMSO | 10 | 99 | 17,300 | 17,500 | 1.05 |
| 6 | Me₆TREN | 1/3 | 0.125 | DMSO | 12 | 99 | 17,300 | 18,200 | 1.07 |
| 7 | Me₆TREN[a] | 1/3 | 0.5 | DMSO | 12 | 99 | 43,300 | 43,600 | 1.09 |
| 8 | Me₆TREN | 1/3 | 0.5-in dark | DMSO | 12 | 0 | – | – | – |
| 9 | Me₆TREN | 1/3 | 0 | DMSO | 12 | 0 | – | – | – |
| 10 | Me₆TREN | 1/5 | 2 | MeCN | 24 | 94 | 16,400 | 15,700 | 1.06 |
| 11 | Me₆TREN | 1/5 | 0 | MeCN | 24 | 0 | – | – | – |
| 12 | Me₆TREN | 1/5 | 2 | DMF | 20 | 99 | 17,300 | 17,000 | 1.05 |
| 13 | Me₆TREN | 1/5 | 0 | DMF | 20 | 0 | – | – | – |
| 14 | TPMA[b] | 1/5 | 0.5 | DMSO | 40 | 93 | 16,200 | 15,600 | 1.07 |
| 15 | TPMA[b] | 1/5 | 0-in dark | DMSO | 24 | 0 | – | – | – |
| 16 | PMDETA | 1/5 | 1 | DMSO | 36 | 90 | 15,700 | 15,500 | 1.12 |
| 17 | PMDETA | 1/5 | 0 | DMSO | 36 | 0 | – | – | – |
| 18 | Me₆TREN | 0.125/3 | 0.25 | DMSO | 5 | 99 | 17,300 | 17,200 | 1.20 |
| 19 | Me₆TREN | 0.25/3 | 0.25 | DMSO | 5 | 99 | 17,300 | 17,100 | 1.10 |
| 20 | Me₆TREN | 0.5/3 | 0.25 | DMSO | 5 | 99 | 17,300 | 17,900 | 1.10 |
| 21 | Me₆TREN | 0.125/3 | 0.125 | DMSO | 12 | 99 | 17,300 | 18,200 | 1.07 |

Polymerizations were conducted in different solvents (50 vol %) and irradiated under green light (0.9 mW/cm²). In entry 1–17, [MA]/[EBiB]/[CuBr₂]/[L] = 200/1/0.04/x (L = Me₆TREN, TPMA, or PMDETA; x = 0.04, 0.08, 0.12, 0.16, or 0.2). In entry 18–21, [MA]/[EBiB]/[CuBr₂]/[Me₆TREN] = 200/1/y/0.12 (y = 0.005, 0.01, and 0.02 corresponding to 25, 50, and 100 ppm with respect to the monomer).
MA methyl acrylate, Conv conversion, Đ = $M_w/M_n$, Me₆TREN tris[2-(dimethylamino)ethyl]amine, TPMA tris(2-pyridylmethyl)amine, PMDETA N,N,N',N'',N''-pentamethyldiethylenetriamine, DMSO dimethylsulfoxide, DMF N,N-dimethylformamide, MeCN acetonitrile, EBiB ethyl α-bromoisobutyrate, DP degree of polymerization.
[a]DP = 500.
[b]Triethanolamine (0.6 equiv. relative to EBiB) was used as the electron donor in the presence of TPMA.

that indicated the presence of mesopores. Scanning electron microscopy (SEM) and transmission electron microscope (TEM) study revealed that the photocatalyst exhibited rough surface and fused, mixed morphology (Supplementary Figs. 6 and 7).

The ultraviolet/visible (UV/Vis) absorption and photoluminescent properties of the PPh₃-CHCP were investigated prior to their applications in photo-ATRP. Solid-state diffused reflectance UV–Vis spectra of PPh₃-CHCP showed a strong absorption over a wide range (200–900 nm), indicating its potential as a broadband light harvester (Fig. 2e). The band gap energy of the photocatalyst was calculated to be $E_g = 1.77$ eV corresponding to ~700 nm (Supplementary Fig. 8). PPh₃-CHCP exhibited a low quantum yield ($\Phi_F = 0.117$, $\lambda_{em} = 580$ nm) and nanoseconds fluorescence lifetime (3.76 ns, $\lambda_{em} = 580$ nm) respectively (Supplementary Fig. 9 and Supplementary Table 1). Meanwhile, photocurrent measurements suggested the charge transfer and separation ability under visible light irradiation (Fig. 2f). As such, PPh₃-CHCP could be suitable to serve as a photocatalyst for photo-ATRP, as it exhibited a wide range of absorption, relatively low photoluminescence quantum yield, and suitable photoelectric response.

**Cu-Catalyzed ATRP using PPh₃-CHCP as photocatalyst under green light irradiation**

PPh₃-CHCP was first evaluated as a photocatalyst under green light irradiation (Supplementary Fig. 10) using methyl acrylate (MA) as a typical monomer, as the absorption of copper complex was weak in this spectrum and most of the side effects were ruled out under these conditions[52]. Polymerizations were conducted using CuBr₂/tris[2-(dimethylamino)ethyl]amine (Me₆TREN) in the presence of PPh₃-CHCP

at room temperature using ethyl α-bromoisobutyrate (EBiB) as an initiator. As shown in Table 1, MA could be successfully polymerized under green light irradiation (0.9 mW/cm²) in the presence of PPh₃-CHCP. Control experiments suggested that no polymerization occurred in the absence of PPh₃-CHCP or in the dark, signifying the importance of the photocatalyst for the regeneration of activating species (Table 1). As TPP (the building block of PPh₃-CHCP) also showed an absorption profile below 350 nm (Supplementary Fig. 11), it was used as a control photocatalyst in initial evaluation. In contrast to PPh₃-CHCP catalyzed polymerization, no PMA was obtained using TPP, which indicated that the conjugated nature of the catalyst may play an important role in providing photocatalytic activity.

PPh₃-HCP was also applied in photo-catalyzed ATRP using similar conditions to PPh₃-CHCP. Results indicated that PPh₃-HCP could only offer PMA with low conversion (19%) (Supplementary Table 2), while PPh₃-CHCP resulted in high conversion (Table 1). As shown in Supplementary Fig. 12 and Supplementary Table 3, the conductivity of PPh₃-CHCP was twofold than that of PPh₃-HCP, suggesting that ion intercalation played an important role to improve polymerization efficiency. Normally, doping of ions enhances the electrical conductivity and improves the dispersity in solvents (Supplementary Fig. 13), which may contribute to polymerizations when high monomer conversion and good control of polymerization were targeted.

It was also observed that no polymerization happened when CuBr₂/Me₆TREN ratio was 1/1, which indicated that an excess of amine electron donor (Me₆TREN) was needed to initiate the polymerization. As shown in Table 1, increasing the concentration of ligand resulted in near-quantitative monomer conversions, yielding well-defined polymers with low dispersity (Đ < 1.1) and controlled molecular weights

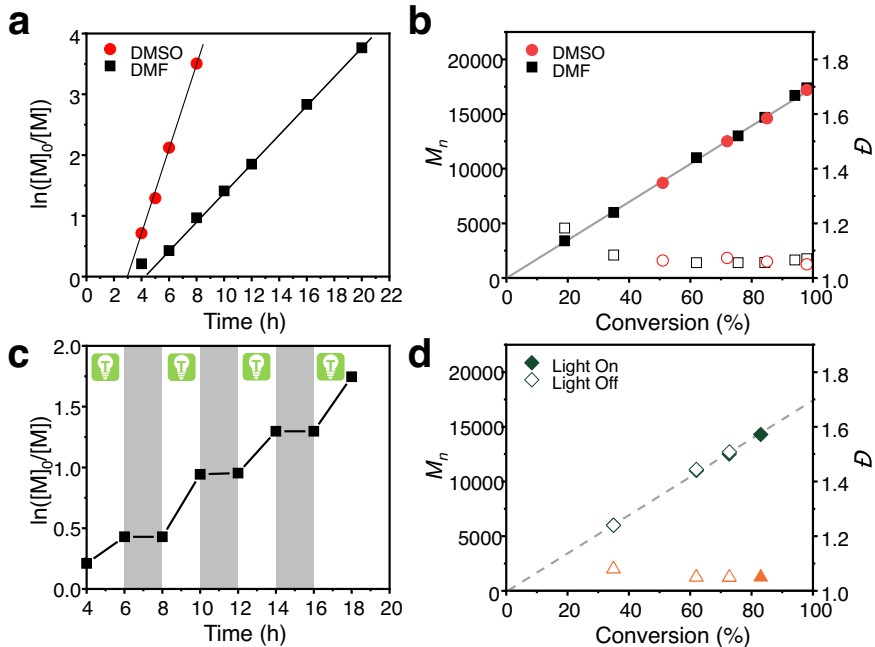

**Fig. 3 | Results for the polymerization of MA using PPh₃-CHCP. a** Kinetics and **b** evolution of molecular weight ($M_n$, filled points) and dispersity ($Đ$, empty points) of the polymers as a function of monomer conversion in ATRP of MA using 0.5 (in DMSO) and 2 mg/mL (in DMF) PPh₃-CHCP photocatalyst with Me₆TREN ligands respectively. **c** Temporal control in ATRP of MA upon intermittent switching green light on/off in the presence of the PPh₃-CHCP photocatalyst. **d** Plot of $M_n$ and $Đ$ (solid symbols indicate after irradiation, and open symbols indicate after dark period) versus monomer conversions, using PPh₃-CHCP as the photocatalyst in DMF during pulsed light irradiation. Reaction conditions: [MA]/[EBiB]/[CuBr₂]/[Me₆TREN] = 200/1/0.04/$x$, $x$ = 0.12 or 0.2 in 50 vol% DMSO or DMF respectively, irradiated under green LEDs (0.9 mW/cm²).

(Table 1, entry 1–4). Related size exclusion chromatography (SEC) results were shown in Supplementary Fig. 14, which exhibited monomodal and symmetric curves. Meanwhile, when concentration of PPh₃-CHCP was decreased from 0.5 to 0.25 and 0.125 mg/mL (corresponding to 0.1, 0.05, and 0.025 wt% with respect to MA, respectively), efficient and well-controlled polymerizations could still be obtained with near-quantitative monomer conversion with prolonged time (Table 1). All obtained polymers exhibited low dispersity and their SEC monomodal results are shown in Supplementary Fig. 14. Polymerization of MA was also performed using tris(2-pyridylmethyl)amine (TPMA) and N,N,N′,N″,N″-pentamethyldiethylenetriamine (PMDETA) as ligands. The monomer conversions reached 93% and 90% (with dispersity of 1.07 and 1.12) from TPMA and PMDETA, respectively (Table 1 and Supplementary Fig. 15), which suggested the high versatility of this system. In addition, polymerization was also successfully conducted using N, N-dimethylformamide (DMF) or acetonitrile (MeCN) as solvent in the presence of PPh₃-CHCP, reaching high monomer conversions and yielding polymers with low dispersity values ($Đ$ < 1.1) and controlled molecular weights (Table 1 and Supplementary Fig. 15). Decreasing concentration of CuBr₂ resulted in an increase in the rate of polymerization, but produced polymers with relatively broad dispersity. For example, fast polymerization of MA was achieved that offered 99% monomer conversion within <5 h but with a high dispersity value of 1.20 (entry 18, Table 1), by reducing the amount of CuBr₂ from 0.04 to 0.005 equiv (with respect to initiator or 25 ppm with respect to monomer) (Supplementary Fig. 16). In the presence of 0.01 or 0.02 equiv (50 or 100 ppm, respectively) of CuBr₂ (entries 19 and 20, Table 1), the resultant polymers showed a low dispersity of 1.10 (Supplementary Fig. 16). Furthermore, the polymerization of MA under both low concentration of PPh₃-CHCP (0.125 mg/mL) and copper catalyst (CuBr₂/Me₆TREN, 25 ppm) (entry 21, Table 1) yielded PMA with a low dispersity value of 1.07 (Supplementary Fig. 17).

Kinetic study was performed in dimethyl sulfoxide (DMSO) and DMF respectively, and results are shown in Fig. 3. PPh₃-CHCP in DMSO

exhibited an induction period (<4 h), after which the polymerization progressed with an apparent rate constant of $k_p^{app}$ = 0.69 h⁻¹, much faster than that observed in DMF ($k_p^{app}$ = 0.24 h⁻¹) (Fig. 3a). Both polymerizations presented a living nature. Molecular weights of obtained polymers were in line with the theoretical values, and their dispersity were low (Fig. 3b and Supplementary Fig. 18). The induction period in the polymerization may be due to the presence of dissolved oxygen in the reaction solutions[45]. In addition, as typical photocatalyzed ATRPs, the polymerizations exhibited a temporal control behavior in response to switching the light on/off (Fig. 3c, d and Supplementary Fig. 19). The dark period was prolonged up to 12 h and related results are shown in Supplementary Fig. 20. A negligible monomer conversion was observed in dark, which suggest that active radicals are not efficiently generated under dark conditions. The polymerizations could be impacted by light intensity. Results of polymerizations under green light with varied light intensities using MA as typical monomer are shown in Supplementary Fig. 21. Remarkable increase of polymerization rate and slight decrease of the induction period in the early stage were observed when the light intensity was increased progressively from 0.6 to 1.5 mW/cm², which is also mentioned in previous reports of photo-RDRP[53].

Kinetic study suggested that PPh₃-CHCP could be applied as a highly efficient photocatalyst. In the PPh₃-CHCP catalyzed photo-ATRP process (shown in Fig. 1b), as a typical PET-ATRP, excitation of the photocatalyst resulted in the separation of electron-hole charge carriers that could generate activating species upon reacting with the Cu catalyst and electron donors, respectively. However, the ATRP activator (L/Cuᴵ) could be generated by an electron transfer from the photocatalyst reducing L/Cuᴵᴵ-Br to L/Cuᴵ. Furthermore, transfer of holes to amine electron donors (D) formed amine radical cation (D⁺·) species and regenerated the initial ground state PPh₃-CHCP photocatalyst. Deprotonation of the amine radical cation may further proceed to generate α-aminoalkyl radicals (D·) that reduce L/Cuᴵᴵ-Br to generate the activator L/Cuᴵ catalyst. Therefore, activating/initiating

species were formed through different pathways in the presence of the photocatalyst to start the ATRP process in this system[54].

Photoexcitation of PPh$_3$-CHCP resulted in the population of its excited state PPh$_3$-CHCP*. Electron transfer between PPh$_3$-CHCP* and alkyl bromides resulted in generation of the propagating radicals. Meanwhile, the oxidized PPh$_3$-CHCP (PPh$_3$-CHCP$^{•+}$) more likely reacted with amine electron donors to regenerate the PPh$_3$-CHCP photocatalyst. This was in some way similar to the organocatalyzed ATRP mechanism reported previously[55]. As a control, irradiating solutions of PPh$_3$-CHCP in DMSO with ethyl α-bromophenylacetate (EBPA) and EBiB were also performed to investigate whether radicals are generated from initiator and PPh$_3$-CHCP (Supplementary Fig. 22). As expected, dimer and Br$_2$ were observed during ESI-mass spectrometric analysis (Supplementary Fig. 23), indicating the significant homolytic cleavage of C-Br bond in the presence of the PPh$_3$-CHCP and light irradiation.

As a heterogeneous photocatalyst, PPh$_3$-CHCP could be easily separated from the reaction mixture and reused in multiple ATRP cycles while retaining its high photocatalytic efficiency. The recycled PPh$_3$-CHCP photocatalyst enabled successive ATRP of MA in the presence of CuBr$_2$/Me$_6$TREN in DMSO. After 5 cycles, near quantitative monomer conversions were still obtained with well-controlled molecular weights and low dispersity values of the polymers (Supplementary Figs. 24 and 25). The recovered photocatalyst was also characterized by SEM-EDS elemental mapping (Supplementary Fig. 26), and no obvious Cu signal was observed suggesting that the photocatalyst has weak ability to trap Cu atoms, which would not impact results in the recycling experiments.

PPh$_3$-CHCP catalyzed ATRP can also be applied to a variety of acrylates with high conversions and yields, together with controlled molecular weights and low dispersity values (Supplementary Table 4). For example, conversion of n-butyl acrylate (BA), (2-methoxyethyl) acrylate (MEA), 2,2,2-trifluoroethyl acrylate (TFEA), benzyl acrylate (BzA), and tert-butyl acrylate (t-BA) was 99%, 99%, 83%, 99%, and 99%, with dispersity of 1.11, 1.08, 1.06, 1.08 and 1.12, respectively. Polymerizations of methyl acrylates were also successfully performed and the related results and SEC curves are shown in Supplementary Fig. 27. Conversion of methyl methacrylate (MMA), n-butyl methacrylate (n-BMA), cyclohexyl methacrylate (CHMA), and lauryl methacrylate (LMA), and poly(ethylene glycol) methyl ether methacrylate (OEGMA) approached 99% with dispersity of 1.12, 1.18, 1.20, 1.12, and 1.29, respectively, among which the polymerization of OEGMA was conducted in NaBr aqueous solution. Polymerization of styrene was also conducted using PPh$_3$-CHCP as photocatalyst under blue light irradiation and achieved 48% monomer conversion (designed DP:100) with relatively low dispersity (1.15) in 24 h (Supplementary Fig. 27 and Supplementary Table 4), which was comparable to the most efficient photoinduced Cu-ATRP system[56].

Chain extension experiments revealed the high living nature of the resulting polymers to enable the formation of di-block copolymers. A PMA macroinitiator (conversion ~99%, $M_n$ = 17700, Đ = 1.06) was applied to initiate the second monomer (MEA) that resulted in 99% conversion and offered a well-defined di-block copolymer ($M_n$ = 42,700, Đ = 1.06) (Supplementary Fig. 27). Meanwhile, PMA ($M_n$ = 8200, Đ = 1.06) obtained from green light irradiation was characterized by $^1$H NMR (Supplementary Fig. 28). The ratio of integral value between methyl group (g, from residual of initiator) and methine group (c, chain end) was 0.96:6, which was very close to the theoretical ratio (1:6), suggesting a good retention of chain-end functionality[57].

## ATRP under broadband light and sunlight

PPh$_3$-CHCP exhibited strong catalytic properties under broadband light irradiation. Typical results from polymerizations of MA and MMA are shown in Supplementary Table 5. Both MA and MMA achieved nearly quantitative conversions under blue (0.9 mW/cm$^2$), orange (2 mW/cm$^2$), red (2 mW/cm$^2$), and white light (0.9 mW/cm$^2$). Performing photocatalyzed RDRP under long wavelength light can suppress side reactions and enhance photon penetration depth in reaction media[49]. The development of dual photoredox catalytic systems with Cu-ATRPs such as the use of zinc(II) tetraphenylporphine (ZnPor) photocatalyst offer opportunities to produce polymers with high yielding under long wavelength light irradiation[44,58]. Therefore, developing suitable photocatalysts for dual photoredox catalytic systems could greatly help to achieve NIR light-induced Cu-ATRP[56]. As PPh$_3$-CHCP exhibited relatively strong absorption in near infra-red (NIR) region, the light wavelength could be extended to NIR (730–940 nm with varying intensities) (Fig. 2e). The reaction time to reach maximum monomer conversion gradually became longer, which may be because the light absorption of PPh$_3$-CHCP decreased with increasing wavelength. The dispersity of the acrylates and methyl acrylates were lesser than 1.09 and 1.12 respectively, which indicated good control over the polymerizations (Supplementary Table 5). Related SEC results showed monomodal and symmetric curves (Fig. 4) and the polymerization rates of PPh$_3$-CHCP system were high. Meanwhile, the irradiation intensity in our system was generally much lower than that of similar systems (Supplementary Fig. 29), and this was beneficial for production within low light intensity conditions such as cloudy day or low output power. High chain-end fidelity could remain under broadband light irradiation according to $^1$H NMR analysis of polymer obtained from blue, red and 940 nm light irradiation (Supplementary Table 6 and Supplementary Fig. 30), respectively. These results indicate the living nature of the photoinduced Cu-ATRPs under light irradiation with different wavelengths.

NIR radiations have remarkable penetrating ability with minimal absorption and scattering. Taking advantage of this feature, we also performed polymerizations in reaction vessels wrapped with opaque paper (thickness ~0.4 mm) as light barriers, achieving efficient monomer conversions (>94%) and in close agreement with the theoretical and experimental molecular weight (Supplementary Table 7 and Supplementary Fig. 31). In conclusion, polymerization rate of NIR light-induced Cu-ATRP using PPh$_3$-CHCP was closer to most of the photocatalyzed RDRPs under NIR light irradiation[37,59] (Supplementary Table 8). Meanwhile, the irradiation intensity in our system was generally much lower and monomer conversions were relatively higher (e.g., 99%). The performance of NIR penetration photo-ATRP using PPh$_3$-CHCP was comparable to the previously reports (Supplementary Table 9). The deep penetration of NIR light offers opportunities to apply large-scale PPh$_3$-CHCP catalyzed Cu-ATRPs (including both batch and microfluidic devices), which is expected to promote scalable production of well-defined polymers.

It would be very valuable for industrial applications to develop solar-driven photopolymerization protocols, if the polymerization can achieve full monomer conversions within a single sunlight irradiation period (e.g., 6–8 h during daytime including cloudy weather) at large scale. Such development largely relies on the synthesis and application of highly efficient photocatalysts, which can be applied persistently. Detailed investigations suggested that PPh$_3$-CHCP could fulfill these requirements. Polymerizations of MA or MMA were achieved within 6 h using 0.5 mg/mL PPh$_3$-CHCP under natural sunlight, resulting in monomer conversions of 99% (Supplementary Table 5). The obtained polymers showed $M_n$ close to the theoretical values and low dispersity values (1.06 and 1.12 for PMA and PMMA respectively, entry 11 and 23 in Supplementary Table 5). In comparison, limited (~48%) or negligible (~0%) monomer conversions were observed for MA and MMA without PPh$_3$-CHCP respectively (Supplementary Table 10 and Supplementary Fig. 32). Considering there was negligible conversion of MA under blue and green light irradiation (Supplementary Table 11), the concentration of Cu$^I$ activator regenerated by UV rays from sunlight was not sufficient to sustain significant chain growth in a short period[60].

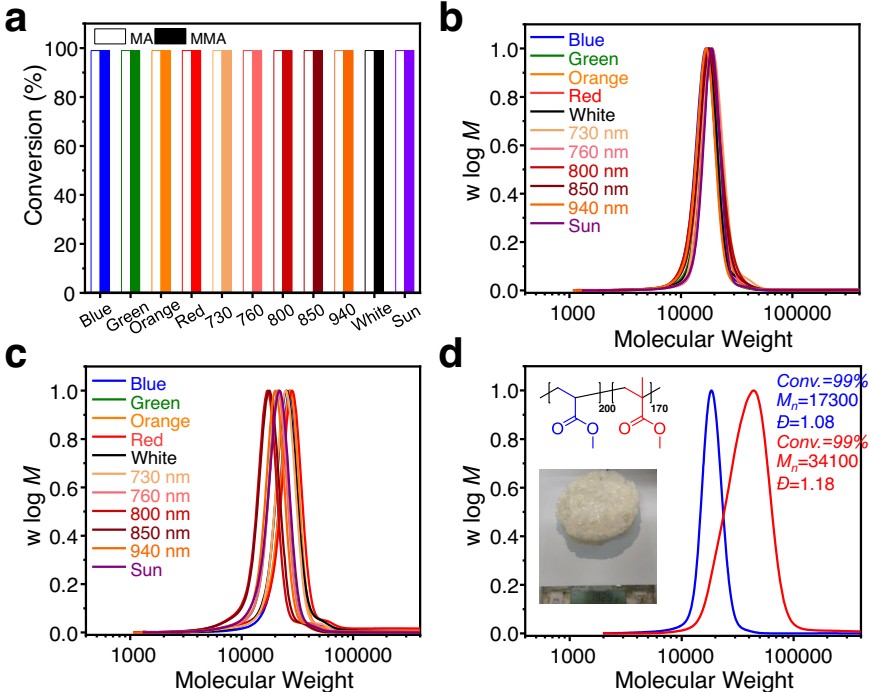

**Fig. 4 | Results for the polymerization of MA and MMA using PPh₃-CHCP under broadband light and sunlight respectively. a** Monomer conversions of MA and MMA using PPh₃-CHCP as photocatalyst under blue, green, orange, red, white, 730 nm, 760 nm, 800 nm, 850 nm, 940 nm, and sunlight irradiation respectively. Experimental details were provided in the supplementary materials. SEC traces of synthesized **b** PMA (color coded) and **c** PMMA (color coded) using PPh₃-CHCP under series light irradiation. **d** SEC traces of 200 mL scale of PMA macroinitiator (in blue) and 400 mL scale of PMA₂₀₀-*b*-PMMA₁₇₀ copolymer (in red, and 192.9 g block copolymer was obtained finally) upon in situ chain extension showing high chain-end fidelity and successful chain extension. The polymers were synthesized using PPh₃-CHCP in the absence of external deoxygenation.

PPh₃-CHCP catalyzed polymerization also demonstrated limited oxygen tolerance. This was evaluated by performing the polymerization of MMA and MA without degassing the reaction mixtures under blue light and sunlight respectively. The polymerizations of MA and MMA were conducted in a sealed but non-degassed vessel (5 mL) using a total liquid volume of 4.8 mL (50/50 (v/v) of solvent/monomer) under blue light irradiation. ¹H NMR characterization revealed a monomer conversion of 99% for MA and MMA after 3.5 h and 6 h, respectively, while SEC showed the presence of PMA ($M_n = 18,800$, Đ = 1.05) and PMMA ($M_n = 19,200$, Đ = 1.13) with good control of molecular weight, which is in good agreement with the theoretical values, and narrow dispersity (Supplementary Table 12 and Supplementary Fig. 33a). Similar results were also obtained under sunlight without deoxygenation process (Supplementary Table 12 and Supplementary Fig. 33b). Limited oxygen tolerance will simplify reactor design and show great practical application potential.

The applications of sunlight-driven photoreaction systems in reported photo-RDRPs (including ATRP and PET-RAFT) were not much efficient in large scale synthesis. Due to the limited penetration depth and photo-scattering, most polymerizations resulted in low conversions or less control over the polymerization process. Herein, we report the large-scale polymerization in the presence of PPh₃-CHCP under sunlight even without deoxygenation. For example, 200 mL scale polymerization (containing 100 mL of MMA or MA) was conducted without deoxygenation (Fig. 1c), and the results are shown in Supplementary Table 5. Both MMA and MA approached 99% conversions within 6 h of sunlight irradiation, producing polymers with dispersity of 1.15 and 1.08, respectively. It is worth mentioning that according to the statistics from the past 20 years of meteorological data, the area where the research was conducted remained cloudy for about 70% of the time[31]. Next, after the polymerization of MA (200 mL, $V_{monomer}/V_{solvent} = 1/1$), MMA solution (200 mL, $V_{monomer}/V_{solvent} = 1/1$)

was in situ added to generate the second block in a 500 mL reaction bottle under blue light irradiation. Both polymerizations were conducted without deoxygenation, and well-defined diblock copolymer ($M_n = 34,100$, Đ = 1.18) was obtained with 99% conversion of MMA (Fig. 4d). Meanwhile, diffusion constants of synthesized PMA₂₀₀-*b*-PMMA₁₇₀ were almost constant (Supplementary Fig. 34), suggesting that there were a few homopolymers of PMA or PMMA in the final product. The monomodal SEC curves and diffusion ordered spectroscopy (DOSY) NMR results indicate the formation of block copolymer. This further suggested the high photo-catalytic efficiency of PPh₃-CHCP.

Therefore, we believe that this is the first report of the full conversion of low active monomer (MMA) in sunlight-driven photocatalyzed polymerization within single solar irradiation period. This is also the largest-scale polymerization process (with diameter ~8.6 cm and almost ~0.6 cm thickness of glass vial) reported for photocatalyzed RDRP under sunlight without deoxygenation (Supplementary Fig. 35). The successful large-scale synthesis of both homopolymers and block copolymers demonstrates its great industrial application potential.

## Discussion

In summary, hyper crosslinked polymers constructed by triphenylphosphine (PPh₃-CHCP) was developed, which has been applied as a highly efficient heterogeneous photocatalyst for activate Cu-catalyzed ATRP. Typical acrylates and methyl acrylates could be polymerized in a control nature with low dispersity values, and molecular weights could be well controlled under 200 ppm amount of Cu catalyst. These polymerizations could be conducted over a wide range of 450–940 nm light irradiation. Meanwhile, the light intensities being applied were very low, and most polymerizations could be completed within 12 h at 99% monomer conversions with a good control. The heterogeneous

nature of the photocatalyst allowed easy separation and reuse in multiple cycles with retention of high photocatalytic efficiency. Polymerizations could also be performed under temporal control, as well as using opaque paper (thickness ~0.4 mm) as light barriers. Sunlight-driven Cu-ATRPs were performed from MA and MMA respectively. Near-quantitative monomer conversions could be obtained under a single sunlight irradiation period (6 h), and these polymerizations exhibited limited tolerance to $O_2$ (without deoxygenation procedure). Both homo- and block copolymer could be obtained with low dispersity values. The sunlight-driven homo-polymerizations could be scaled up to 200 mL, which is the largest reaction scale being reported up to present. The scale of chain extension approached 400 mL, which is also the largest reaction scale being reported till now. We envision that this $PPh_3$-CHCP platform will provide a feasible solution for the large-scale sunlight-driven RDRP production with great potential for industrial applications. We also anticipate that tuning the physical and structural properties will lead to the discovery of a series of CHCPs with capabilities for a variety of applications beyond photocatalysis.

## Methods

### Synthesis of $PPh_3$-HCP
$PPh_3$ (400 mg, 1.53 mmol, 1 equiv.), dimethoxybenzene (1.6 g, 11.59 mmol, 7.6 equiv.), and anhydrous $FeCl_3$ (5.6 g, 34.52 mmol, 22.6 equiv.) were charged into a 50 mL 2-neck flask followed by addition of 20 mL nitrobenzene under dry nitrogen atmosphere. The flask was placed in an oil bath and was heated at 80 °C for 4 h, after which the temperature was increased to 120 °C to allow the complete formation of the network for 20 h. After the reaction, the precipitate was extensively washed with MeOH, acetone, and THF until the filtrate became clear. The network was further Soxhlet extracted in MeOH and THF for 24 h and followed by drying under vacuum to yield ~1.89 g of the $PPh_3$-HCP (yield: 95%) in dark brown color.

### Synthesis of $PPh_3$-CHCP
$PPh_3$-HCP (400 mg) and 50 mL MeOH were charged into a 100 mL 2-neck flask followed by the addition of 300 mg $NaBH_4$ under a nitrogen atmosphere. The flask was placed in an oil bath and was heated at 80 °C for 4 h and cooled to room temperature for 4 h. After the reaction, the mixture was separated by centrifugation and the residue was washed with 2.5 M NaOH aqueous solution, deionized water, and MeOH, respectively. Finally, the network was allowed to dry under vacuum to yield ~310 mg (yield: 72.5%) of the $PPh_3$-CHCP.

### General procedure for photoinduced ATRP of acrylates using $PPh_3$-CHCP under green light irradiation
Photoinduced ATRP process of different acrylates using $PPh_3$-CHCP were similar. Typical procedure for ATRP of MA was as follows: the photocatalyst $PPh_3$-CHCP (0.2–0.8 mg), MA (810 μL, 9.0 mmol, 200 equiv.), DMSO (810 μL), and a stock solution of $CuBr_2$ (0.4 mg, 1.8 μmol, 0.04 equiv.), and $Me_6TREN$ (1.44 μL, 5.4 μmol, 0.12 equiv.) in DMSO (20 μL) were added to a Schlenk tube under nitrogen atmosphere. The tube equipped with a magnet bar was sealed with a rubber septum and degassed by three freeze-vacuum-thaw cycles. A 6.6 μL aliquot of EBiB (45.0 μmol, 1 equiv.) was introduced into the tube via syringe. The tube was irradiated under green LEDs to start the polymerization. Samples were taken periodically and analyzed by $^1H$ NMR and SEC to determine the monomer conversion and molecular weight properties, respectively. PMA could be obtained after filtration of $PPh_3$-CHCP followed by precipitation in methanol directly.

### General procedure for NIR photoinduced ATRP of methyl acrylate
NIR photoinduced ATRP process of methyl acrylate using $PPh_3$-CHCP were conducted with varied amount of photocatalyst, monomer, solvent (DMSO), catalyst ($CuBr_2/Me_6TREN$), and initiator (EBiB). Typical

synthesis of PMA was as follows: photocatalyst $PPh_3$-CHCP (1.6 mg), MA (810 μL, 9.0 mmol, 200 equiv.), DMSO (810 μL), and a stock solution of $CuBr_2$ (0.4 mg, 1.8 μmol, 0.04 equiv.), and $Me_6TREN$ (2.4 μL, 9.0 μmol, 0.2 equiv.) in DMSO (20 μL) were added to a Schlenk tube under nitrogen atmosphere. The tube equipped with a magnet bar was sealed with a rubber septum and degassed by three freeze-vacuum-thaw cycles. A 6.6 μL aliquot of EBiB (45.0 μmol, 1 equiv.) was introduced into the tube via syringe. The tube was irradiated under 940 nm (15 mW/cm²) to start the polymerization. Samples were taken periodically and analyzed by $^1H$ NMR and SEC to determine the monomer conversion and molecular weight properties, respectively. PMA can be obtained after filtration of $PPh_3$-CHCP followed by precipitation in methanol directly.

### General procedure for polymerization of MA in blue light and sunlight irradiation without deoxygenation procedure
Typical procedure was as follows: the photocatalyst $PPh_3$-CHCP (2.4 mg), MA (2.4 mL, 27.0 mmol, 200 equiv.), DMSO (2.4 mL), and a stock solution of $CuBr_2$ (1.2 mg, 5.4 μmol, 0.04 equiv.), and $Me_6TREN$ (7.2 μL, 27 μmol, 0.2 equiv.) in DMSO (60 μL) were introduced to 5 mL vial. A 19.8 μL aliquot of EBiB (0.135 mmol, 1 equiv.) was introduced into the tube via syringe. The tube was irradiated under blue LEDs or sunlight to start the polymerization, respectively. Samples were taken and analyzed by $^1H$ NMR and SEC to determine the monomer conversion and molecular weight properties, respectively. PMA could be obtained after filtration of $PPh_3$-CHCP followed by precipitation in methanol directly.

### General procedure for large scale oxygen tolerance polymerization of MA using sunlight, and block copolymerization of MMA using blue light
Typical procedure was as follows: a 250 mL bottle was charged with a stirbar and $PPh_3$-CHCP (100 mg) was transferred into nitrogen-atmosphere glovebox. DMSO (100 mL), MA (100 mL, 1.11 mol, 200 equiv.), EBIB (813 μL, 5.55 mmol, 1 equiv.) and a stock solution of $CuBr_2$ (49.6 mg, 0.222 mmol, 0.04 equiv.), and $Me_6TREN$ (296 μL, 1.11 mmol, 0.2 equiv.) in DMSO (2 mL) were added sequentially via pipette. The bottle was then removed from the glovebox, and placed on the roof of ShengHua Build (7-floored), Hefei University of Technology for 6 h.

Subsequently, a 500 mL bottle was charged with a stirbar, $PPh_3$-CHCP (100 mg), and the synthesized PMA-Br solution (~200 mL, 5.55 mmol) was transferred into the bottle in glovebox. DMSO (100 mL), MMA (100 mL, 0.955 mol, 170 equiv.), and a stock solution of $CuBr_2$ (49.6 mg, 0.222 mmol, 0.04 equiv.), and $Me_6TREN$ (296 μL, 1.11 mmol, 0.2 equiv.) in DMSO (2 mL) were added sequentially via pipette. The bottle was then removed from the glovebox, and irradiated under blue LEDs (2 mW/cm²) to start the polymerization for 8 h. Block copolymer PMA-$b$-PMMA could be obtained after filtration of $PPh_3$-CHCP followed by precipitation in methanol directly.

## Data availability
All data are available in the main text or the supplementary materials. All other relevant data are available from the corresponding authors upon request. Source data are provided with this paper. Wei-wei, Fang; Yang, Gui yuan; Fan, Zi-Hui; Chen, Zi-Chao; hu, xunliang; Zhang, Zhen; et al. (2023): Large-Scale Photocatalyzed Atom Transfer Radical Polymerization Driven by Sunlight. figshare. Dataset. https://doi.org/10.6084/m9.figshare.22709701.

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

## Acknowledgements

This work was financially supported by National Natural Science Foundation of China (22171067, 22161142005, 21975086), International S & T Cooperation Program of China (2018YFE0117300), and the Science and Technology Department of Hubei Province (No. 2019CFA008).

## Author contributions

T.H., B.E.T., and W.W.F. conceived the idea; T.H. and B.E.T guided the project and wrote the manuscript; W.W.F. and G.Y.Y. performed the experiments; Z.H.F., Z.C.C., X.L.H., Z.Z., I.H., and Y.L. supported and participated the acquisition of data; and T.H., B.E.T., W.W.F., I.H., and Y.L. analyzed the data and participated in the preparation of the manuscript.

## Competing interests

The authors declare no competing interests.
