## [Peer Review File · Nature Communications]

Large-Scale Photocatalyzed Atom Transfer Radical Polymerization Driven by SunlightReviewers' Comments:

Reviewer #1:

Remarks to the Author:

Tan and co-workers report on the development and application of a polymer-based photosensitizer for photo-ATRP. This work appears to have been carefully performed and sufficient details are included to enable reproduction. This work is noteworthy in that a large range of wavelengths can be effectively used to drive the photo-ATRP process. This aspect would facilitate different users of this system to effectively perform successful polymerizations without variances in wavelength. The polymers produced are well-defined and excellent control is demonstrated. This work is suitable for publication in Nature Communications after the authors address the following points or questions.

-Can the authors include structures of the resulting polymer in Figure 1a to help with a diverse audience of readers.

-Is the polymerization impacted by light intensity? This would be a very useful aspect for the authors to include in this work and would make this work more impactful.

-Why is there an induction period in the polymerization, highlighted in Figure 2a?

-Is any copper trapped in the photosensitizer during recycling and could this impact results in the recycling experiments?

Reviewer #2:

Remarks to the Author:

This is a very interesting paper which reports the use of photoATRP for the fabrication of functional polymers. In the recent, photoRDDR has generated lot of interests due to the broad range of applications. The novelty of this paper resides in two points:

- i) the use of phosphine-based conjugated hypercrosslinked polymer as photocatalyst to mediate ATRP.
- ii) the broad range of wavelengths which can be employed to activate this ATRP as well as the direct use of sun light. Indeed, photoRDRP activated by NIR light is still underdeveloped and this paper addresses this. Furthermore, the ability to recycle the catalyst and reuse enables to improve the sustainability of this process. I would like to recommend this paper after some additional work.

The authors should include previous works using MOF photocatalysts which have been successfully employed in PET-RAFT or photoATRP. Indeed, several types of PC have been reported in the recent years.

The authors should demonstrate a broader range of monomers, if possible.

I am not convinced by the chain extension of PMA using MMA as monomer. The authors should include additional proofs to support successful formation of diblock copolymers.

Furthermore, characterizations such as end group characterization should be provided by the authors.

It will be interesting to compare the effect of different wavelengths on the end group.

The performance of this system using NIR should be compared to other NIR activated polymerization, including PET-RAFT. Indeed, several PCs have been developed in the recent years to control photoRDRP under NIR light. The experiment with the paper as barrier is interesting, but this should be compared to previous PCs reported using NIR.

Some of the characterization of the hyperbranched polymers in SI could be moved in main text.

Reviewer #3:

Remarks to the Author:

This manuscript describes the development of crosslinked polymers (PPh₃-CHCP) from PPh₃ and p-dimethoxybenzene, which can be used as an effective heterogeneous photocatalyst to activate the Cu-catalyzed ATRP. This photo and copper co-catalytic system showed a high efficiency under the

irradiation of a wide range of light (450-940 nm). Polymerization at higher volumes (200-400 mL) was also successfully demonstrated. The high efficiency under NIR light is quite impressive. Overall, this work could be suitable for publication after addressing the following concerns:

- The known development of heterogeneous photocatalysts via conjugated cross-linked polymers for activating Cu-catalyzed ATRP (e.g. Ref. 41) as well as examples of ATRP under red light (e.g. ACS Macro Lett. 2022, 11, 3, 376) should be mentioned and discussed.
- ‘... has remained a big challenge. There was also no report on large scale photo-induced production of block copolymers’ The tone of this sentence should be attenuated a little bit, as some known catalytic systems (e.g. PET-RAFT polymerizations) as well as flow chemistry can be applied to a larger scale synthesis.
- In Fig. 1 b, the TOC graphic, as well as several places in the main text, the copper (Cu) catalysts should be mentioned with PPh3-CHCP together, otherwise, could be a little bit misleading and give the readers a feeling that the ATRP was catalyzed by PPh3-CHCP alone without Cu/ Me6TREN.
- Supplementary Table 6, without PPh3-CHCP, 48% conversion of MA was also observed. The authors should comment on this result and perform more control experiments with MA under blue and green light.
- In Fig. 2, C, the results showed no apparent conversion in the dark periods. How about a prolonged dark period (5 h or 12 h) after initiation. The best is also to provide a plot of conversion over time in the Supplementary Information.
- The light sources (emission spectra) as well as details about the ATRP experiment under NIR should be provided in the Supplementary Information.

Given below are our point-by-point responses (in BLUE colour) to the reviewers' comments. The changes to the manuscript and supplementary information are marked in RED colour.

Reviewer 1

Tan and co-workers report on the development and application of a polymer-based photosensitizer for photo-ATRP. This work appears to have been carefully performed and sufficient details are included to enable reproduction. This work is noteworthy in that a large range of wavelengths can be effectively used to drive the photo-ATRP process. This aspect would facilitate different users of this system to effectively perform successful polymerizations without variances in wavelength. The polymers produced are well-defined and excellent control is demonstrated. This work is suitable for publication in Nature Communications after the authors address the following points or questions.

Response: We appreciate the positive comments and suggestions that helped us to significantly improve the manuscript.

1. Can the authors include structures of the resulting polymer in Figure 1a to help with a diverse audience of readers.

Response: Thank you very much for your valuable suggestion. We are sorry for missing the structures of PPh₃-CHCP in initial submission. Due to the amorphous and disordered structure as well as the possible carbonization during the construction of polymer networks (*Nat. Energy* **4**, 604-611 (2019)), the structure of CHCP is complicated and difficult to determine exactly. However, following the previous report (*J. Am. Chem. Soc.* **143**, 9630-9638 (2021)), we have proposed the possible polymer structure in Supplementary Figure 1 for your reference.

Supplementary Figure 1. A proposed structure of the $PPh_3\text{-CHCP}$.

2. Is the polymerization impacted by light intensity? This would be a very useful aspect for the authors to include in this work and would make this work more impactful.

Response: Thank you very much for your valuable suggestion. As suggested, we conducted polymerizations under green light with varied light intensities using methyl acrylate (MA) as a typical monomer. Results indicated that increase of polymerization rate and slight decrease of the induction period in the early stage were observed when the light intensity was increased progressively from 0.6 to 1.5 mW/cm^2 (Supplementary Figure 21) respectively. This was also mentioned in previous photo-RDRP reports (*ACS Macro Lett.* **2**, 592-596 (2013); *Macromolecules* **46**, 2576-2582 (2013); *Angew. Chem. Int. Ed.* **58**, 5170-5189 (2019)).

Accordingly, we revised the manuscript as follows:

The polymerizations could be impacted by light intensity. Results of polymerizations under green light with varied light intensities using MA as typical monomer are shown in Supplementary Fig. 21. Remarkable increase of polymerization rate and slight decrease of the induction period in the early stage were observed when the light intensity was increased progressively from 0.6 to 1.5 mW/cm^2 , which is also mentioned in previous reports of photo-RDRP⁵².

Supplementary Figure 21. (a) Kinetics and (b) evolution of molecular weight (M_n , filled points) and dispersity (\bar{D} , empty points) of the polymers as a function of monomer conversion in the photo-ATRP of MA using 0.5 mg/mL with PPh₃-CHCP as photocatalyst under green light irradiation with different light intensities. Reaction conditions: [MA]/[EBiB]/[CuBr₂]/[Me₆TREN] = 200/1/0.04/0.12 in 50 vol% DMSO.

Reference

52. Corrigan, N. Yeow, J. Judzewitsch, P. Xu, J. & Boyer, C. Seeing the light: advancing materials chemistry through photopolymerization. *Angew. Chem. Int. Ed.* **58**, 5170-5189 (2019).

3. Why is there an induction period in the polymerization, highlighted in Figure 2a?

Response: Typical heterogenous photocatalyzed RDRP systems including photo-ATRP, PET-RAFT and PET-RCMP, have an induction period (*Green Chem.* **18**, 1475-1481 (2016); *J. Am. Chem. Soc.* **143**, 9630-9638 (2021); *Angew. Chem., Int. Ed.* **57**, 12037-12042 (2018); *Angew. Chem. Int. Ed.* **58**, 12096-12101 (2019); *Angew. Chem. Int. Ed.* **61**, e202208898 (2022)), and this may partially be due to the presence of dissolved oxygen in the reaction solutions (*Angew. Chem., Int. Ed.* **57**, 12037-12042 (2018)).

Accordingly, we have revised the manuscript as follows:

The induction period in the polymerization may be due to the presence of dissolved oxygen in the reaction solutions⁴⁵.

Reference

45. Jiang, J. Ye, G. Wang, Z. Lu, Y. Chen, J. & Matyjaszewski, K. Heteroatom-doped carbon dots (CDs) as a class of metal-free photocatalysts for PET-RAFT polymerization under visible light and sunlight. *Angew. Chem., Int. Ed.* **57**, 12037-12042 (2018).

4. Is any copper trapped in the photosensitizer during recycling and could this impact results in the recycling experiments?

Response: Thank you very much for your valuable comment. As suggested, the photocatalyst was recovered and characterized by scanning electron microscopy-energy dispersive spectroscopy (SEM-EDS) for elemental mapping (Supplementary Figure 26). No obvious Cu signal was observed, suggesting that the photosensitizer has weak ability to trap Cu atoms. This indicated that the recycled photocatalyst would not impact results in the recycling experiments.

We also revised the manuscript as follows:

The recovered photocatalyst was also characterized by SEM-EDS elemental mapping (Supplementary Fig. 26), and no obvious Cu signal was observed suggesting that the photocatalyst has weak ability to trap Cu atoms, which would not impact results in the recycling experiments.

Supplementary Figure 26. Elemental mapping of recycled PPh₃-CHCP determined by SEM-EDS.

Reviewer 2

This is a very interesting paper which reports the use of photoATRP for the fabrication of functional polymers. In the recent, photoRDDR has generated lot of interests due to the broad range of applications. The novelty of this paper resides in two points:

- 1) the use of phosphine-based conjugated hypercrosslinked polymer as photocatalyst to mediate ATRP.
- ii) the broad range of wavelengths which can be employed to activate this ATRP as well as the direct use of sun light. Indeed, photoRDRP activated by NIR light is still underdeveloped and this paper addresses this. Furthermore, the ability to recycle the catalyst and reuse enables to improve the sustainability of this process. I would like to recommend this paper after some additional work.

Response: We appreciate the positive comments and suggestions that helped us to significantly improve the manuscript.

- 1) The authors should include previous works using MOF photocatalysts which have been successfully employed in PET-RAFT or photoATRP. Indeed, several types of PC have been reported in the recent years.

Response: Thank you very much for your valuable suggestion, We are sorry for missing out important literature in the initial submission to keep the manuscript within the recommended words limit. MOFs with various functional building blocks and dimensions have been used as highly efficient heterogenous photocatalysts for photo RDRPs (*Angew. Chem. Int. Ed.* **60**, 5489-5496 (2021); *Angew. Chem. Int. Ed.* **60**, 22664-22671 (2021)). For example, porphyrinic zirconium MOF (MOF-525(Zn)) and two dimension (2D) ZnTCPP nanosheets (TCPP: 5,10,15,20-(tetra-4-carboxyphenyl) porphyrin) were applied as photocatalysts in stereolithographic 3D printing, and created 3D polymeric objects using visible light irradiation for a short period.

Accordingly, we revised the manuscript as follows:

Metal organic frameworks (MOFs) with various functional building blocks and dimension such as porphyrinic zirconium MOF (MOF-525(Zn)) and two dimensional (2D) ZnTCPP nanosheets (TCPP: 5,10,15,20-(tetra-4-carboxyphenyl) porphyrin), have

been employed as highly efficient heterogeneous photocatalysts in stereolithographic 3D printing, and created 3D polymeric objects using visible light irradiation in short period^{46,47}. However, sunlight was not applied in these polymerizations.

Reference

46. Zhang, L. et al. 2D porphyrinic metal-organic framework nanosheets as multidimensional photocatalysts for functional materials. *Angew. Chem. Int. Ed.* **60**, 22664-22671 (2021).

47. Zhang, L. et al. Porphyrinic zirconium metal-organic frameworks (MOFs) as heterogeneous photocatalysts for PET-RAFT polymerization and stereolithography. *Angew. Chem. Int. Ed.* **60**, 5489-5496 (2021).

2) The authors should demonstrate a broader range of monomers, if possible.

Response: Thank you very much for your valuable suggestion. Actually, various monomers including methacrylates and methyl methacrylates, etc. can be applied in our photo-ATRPs. Keeping in view the scope of this submission, we applied tert-butyl acrylate (t-BA), cyclohexyl methacrylate (CHMA), and lauryl methacrylate (LMA) as additional monomers in photo-ATRPs. Conversion of t-BA, CHMA, and LMA was 99%, 98% and 99% with dispersity of 1.12, 1.20, and 1.12, respectively. Related results were shown in Supplementary Table 4 and Supplementary Fig. 27 respectively.

We also revised the manuscript as follows:

For example, conversion of n-butyl acrylate (BA), (2-methoxyethyl) acrylate (MEA), 2,2,2-trifluoroethyl acrylate (TFEA), benzyl acrylate (BzA), and tert-butyl acrylate (t-BA) was 99%, 99%, 83%, 99%, and 99%, with dispersity of 1.11, 1.08, 1.06, 1.08 and 1.12, respectively. Polymerizations of methyl acrylates were also successfully performed and the related results and SEC curves are shown in Supplementary Fig. 27. Conversion of methyl methacrylate (MMA), n-butyl methacrylate (n-BMA), cyclohexyl methacrylate (CHMA), and lauryl methacrylate (LMA), and poly(ethylene glycol) methyl ether methacrylate (OEGMA) approached 99% with dispersity of 1.12, 1.18, 1.20, 1.12, and 1.29, respectively, among which the polymerization of OEGMA was conducted in NaBr aqueous solution.

Supplementary Table 4. Monomer scope of PPh₃-CHCP photocatalyzed ATRPs

entry	Monomer	Time (h)	Conv (%)	M _{n,th}	M _n	Đ
1	t-BA	24	99	12000	11400	1.12
2	CHMA	20	98	14400	12600	1.20
3	LMA	24	99	9300	11200	1.12

Reaction condition: [t-BA]/[Br-PEG-Br]/[CuBr₂]/[Me₆TREN] = 60/0.5/0.04/0.2 in 67 vol% MeCN. [CHMA]/[Br-PEG-Br]/[CuBr₂]/[Me₆TREN] = 60/0.5/0.04/0.2 in 67 vol% DMF. [LMA]/[Br-PEG-Br]/[CuBr₂]/[Me₆TREN] = 20/0.5/0.04/0.2 in 67 vol% DMF. PPh₃-CHCP = 2 mg/mL.

Supplementary Figure 27. Results of ATRP of acrylate monomers using PPh₃-CHCP photocatalyst. Experimental details are provided in the Supplementary Table 4.

3) I am not convinced by the chain extension of PMA using MMA as monomer. The authors should include additional proofs to support successful formation of diblock copolymers.

Response: Thank you very much for your valuable suggestion. As molecular weight of block copolymers ($M_n = 34100$) exceeded the range of matrix-assisted laser desorption/ionization time of flight mass spectrometry (MALDI-TOF MS), diffusion ordered spectroscopy (DOSY) NMR was employed to evaluate the formation of the block copolymers (*Polym. Chem.* **7**, 5249-5257 (2016); *Polym. Chem.* **10**, 6447-6455 (2019)). As shown in Supplementary Figure 35, diffusion constants of the block copolymer poly(methacrylate₂₀₀-*b*-methyl methacrylate₁₇₀) were almost constant, suggesting that there were a few homopolymers of PMA or PMMA in the final product. The shift of

monomodal GPC curves and DOSY NMR results supported the successful formation of block copolymer.

We also revised the manuscript as follows:

Meanwhile, diffusion constants of synthesized PMA₂₀₀-*b*-PMMA₁₇₀ were almost constant (Supplementary Fig. 34), suggesting that there were a few homopolymers of PMA or PMMA in the final product. The monomodal GPC curves and diffusion ordered spectroscopy (DOSY) NMR results indicate the formation of block copolymer.

Supplementary Figure 34. DOSY NMR spectrum of PMA₂₀₀-*b*-PMMA₁₇₀.

4. Furthermore, characterizations such as end group characterization should be provided by the authors. It will be interesting to compare the effect of different wavelengths on the end group.

Response: Thank you very much for your valuable suggestions, ¹H NMR could be applied in the chain end analysis of PMAs (*J. Am. Chem. Soc.* 2014, 136, 1141-1149). As shown in Supplementary Figure 28, PMA ($M_n = 8200$, $D = 1.06$) obtained from green light irradiation was characterized by ¹H NMR. The ratio of integral value between methyl group (g, from residual of initiator) and methine group (c, chain end) was 0.96:6, which was very close to the theoretical ratio (1:6). Similar results were obtained from varied PMAs synthesized under blue, red and 940 nm light irradiation respectively (Supplementary Table 6 and Supplementary Figure 30). These results indicated the living nature of the photoinduced Cu-ATRP under light irradiation with different

wavelengths.

We revised the manuscript as follows:

Meanwhile, PMA ($M_n = 8200$, $D = 1.06$) obtained from green light irradiation was characterized by ^1H NMR (Supplementary Fig. 28). The ratio of integral value between methyl group (g, from residual of initiator) and methine group (c, chain end) was 0.96:6, which was very close to the theoretical ratio (1:6), suggesting a good retention of chain-end functionality⁵⁶.

Supplementary Figure 28. ^1H NMR spectrum of PMA obtained from green light irradiation (0.9 mW/cm^2). Integrated ratio of g : c was 0.96 : 6.00. Reaction condition: $[\text{MA}]/[\text{EBiB}]/[\text{CuBr}_2]/[\text{Me}_6\text{TREN}] = 100/1/0.04/0.12$ in 50 vol % DMSO, $\text{PPh}_3\text{-CHCP} = 0.5 \text{ mg/mL}$.

High chain-end fidelity could remain under broadband light irradiation according to ^1H NMR analysis of polymer obtained from blue, red, and 940 nm light irradiation (Supplementary Table 6 and Supplementary Fig. 30) respectively. These results indicated the living nature of the photoinduced Cu-ATRP under light irradiation with different wavelengths.

Supplementary Figure 30. ^1H NMR (CDCl_3 , 600 MHz) spectra of varied PMAs synthesized under blue, green, red and 940 nm light irradiation respectively. g and c represented methyl group (from residual of initiator) and methine group (chain end) respectively. Details were provided in the Supplementary Table 6.

Supplementary Table 6. Results of PMAs synthesized from photo-ATRP under various light irradiation with chain end analysis.

entry	Light Source	Time (h)	Conv (%)	$M_{n,\text{NMR}}$	$M_{n,\text{SEC}}$	\bar{D}	g:c
1	Blue-0.9 mW/cm ²	3	99	7800	7600	1.07	0.99:6
2	Green-0.9 mW/cm ²	10	98	8400	8200	1.06	0.96:6
3	Red-2 mW/cm ²	12	98	7600	8500	1.06	0.99:6
4	940 nm-15 mW/cm ²	12	96	8000	8000	1.06	0.97:6

Reaction condition: $[\text{MA}]/[\text{EBiB}]/[\text{CuBr}_2]/[\text{Me}_6\text{TREN}] = 100/1/0.04/x$ in 50 vol% DMSO ($x=0.12$ and 0.5 mg/mL $\text{PPh}_3\text{-CHCP}$ for blue and green light irradiation, $x=0.2$ and 1 mg/mL $\text{PPh}_3\text{-CHCP}$ for red and 940 nm light irradiation).

Reference

56. Anastasaki, A. et al. Copper (II)/tertiary amine synergy in photoinduced living radical

polymerization: Accelerated synthesis of ω -functional and α, ω -heterofunctional poly (acrylates). *J. Am. Chem. Soc.* **136**, 1141-1149 (2014).

5) The performance of this system using NIR should be compared to other NIR activated polymerization, including PET-RAFT. Indeed, several PCs have been developed in the recent years to control photoRDRP under NIR light. The experiment with the paper as barrier is interesting, but this should be compared to previous PCs reported using NIR.

Response: Thank you very much for your valuable suggestions. According to literature, various photocatalysts including bacteriochlorophyll α (*Angew. Chem.* **128**, 1048-1052 (2016)), zwitter ionic polymethine (*Angew. Chem. Int. Ed.* **57**, 7898-7902 (2018)), aluminium naphthalocyanine (*Macromolecules* **49**, 3274-3285 (2016), *Angew. Chem. Int. Ed.* **59**, 2013-2017 (2020)), metalloporphyrin (*Nat. Commun.* **12**, 478 (2021)), noble metal nanoparticles or their composites (*Angew. Chem. Int. Ed.* **58**, 12096-12101 (2019); *Polym. Chem.* **13**, 1022-1030 (2022)), upconversion nanoparticles or their composites (*Macromolecules* **53**, 4678-4684 (2020); *ACS Appl. Mater. Interfaces* **14**, 21555-21563 (2022)), have been developed for NIR activated RDRPs (Supplementary Table 8). The polymerization rate of our system was close to most of the photocatalyzed RDRPs under NIR irradiation (*Angew. Chem. Int. Ed.* **59**, 2013-2017 (2020); *Nat. Commun.* **12**, 478 (2021)). Meanwhile, the irradiation intensity in our system was generally much lower and monomer conversions were relatively higher (e.g., 99%). In addition, the performance of NIR penetration photo ATRP using PPh₃-CHCP was comparable to the previously reports (Supplementary Table 9). The deep penetration of NIR light offered opportunities to apply large-scale PPh₃-CHCP catalyzed Cu-ATRP (including both batch and microfluidic devices), which is expected to promote scalable production of well-defined polymers.

We revised the manuscript as follows:

In conclusion, polymerization rate of NIR light-induced Cu-ATRP using PPh₃-CHCP was closer to most of the photocatalyzed RDRPs under NIR light irradiation^{37,58} (Supplementary Table 8). Meanwhile, the irradiation intensity in our system was

generally much lower and monomer conversions were relatively higher (e.g., 99%). The performance of NIR penetration photo-ATRP using PPh₃-CHCP was comparable to the previously reports (Supplementary Table 9). The deep penetration of NIR light offers opportunities to apply large-scale PPh₃-CHCP catalyzed Cu-ATRPs (including both batch and microfluidic devices), which is expected to promote scalable production of well-defined polymers.

Supplementary Table 8. Summary of NIR photo RDRPs

Material	Method	Light Source	Monomer	Conv (%)	Time (h)	Ref.
PPh ₃ -CHCP	P-ATRP	940 nm, 15 mW/cm ²	MA	99	12	This Work
			MMA	99	44	
NIR-Dye	P-ATRP	790 nm, 100 mW/cm ²	MMA	59	48	14
UCNP@SiO ₂ @N-CDs	P-ATRP	980 nm, 1.5 W/cm ²	MMA	46	22	21
			HEA	66	5	
β -NaYF ₄ :30% Yb ³⁺ , 1% Tm ³⁺	P-ATRP	980 nm, 4 W/cm ²	MA	67	24	23
			MMA	20		
			AN	60	12	
ZnTtBAzP	PET-RAFT	780 nm, 10 W/cm ²	MA	80	3.3	19
AlNc	PET-RAFT	850 nm, 20 mW/cm ²	MA	82	0.5	8
Bacteriochlorophyll α	PET-RAFT	780 nm, 20 mW/cm ²	MMA	44	16	18
Ag ₃ PO ₄	PET-RAFT	940 nm, 16 mW/cm ²	BzA	99	18.8	5
SA-TCPP	PET-RAFT	850 nm, 4.0 mW/cm ²	DMA	29.8	64	6
AlPc	PET-RAFT	780 nm, 6.2 mW/cm ²	MA	66	6	22
RTPP	PET-RAFT	740 nm, 66 mW/cm ²	MMA	89	12	20
			BA	30	6	
CsPbBr ₃ NCs	PET-RAFT	800 nm, 3 W/cm ²	MA	60.2	11	24
CsPbI ₃ @PCN-222	PET-RAFT	850 nm, 50 mW/cm ²	MMA	99	10	25
NaYF ₄ :Yb/Tm	PET-RAFT	980 nm, 2 W/cm ²	n-BA	100	5	26
β -NaYF ₄ :Yb/Tm	SI-PET-RAFT	980 nm, 1.5 W/cm ²	t-BA	78	24	27
Au NR	PET-RAFT	980 nm, 0.5 mW/cm ²	MMA	16.5	72	28
Au/g-C ₃ N ₄	PET-RAFT	740 nm, 1.5 mW/cm ²	MMA	18.1	20	29

Supplementary Table 9. Summary of NIR penetration photo RDRP with material barriers

Material	Method	Light Source	Monomer	Conv (%)	Time (h)	Barrier	Ref.
PPh ₃ -CHCP	P-ATRP	940 nm, 30 mW/cm ²	MA	94	6	0.4 mm A4 paper	This Work
β - NaYF ₄ :30% Yb ³⁺ , 1% Tm ³⁺	P-ATRP	980 nm, 4 W/cm ²	MMA	64 88	36	0.2 mm A4 paper 1.2 mm pig skin	23
UCNP@Si O ₂ @N-CDs	P-ATRP	980 nm, 1.5 W/cm ²	MMA	45	24	1.2 mm pig skin	21
Bacteriochl orophyll α	PET- RAFT	850 nm, 40 mW/cm ²	MMA	24	20	0.2 mm A4 paper	18
Ag ₃ PO ₄	PET- RAFT	780 nm, 6 mW/cm ²	BzA	88	21	0.1 mm A4 paper	5
AlNc	PET- RAFT	850 nm, 100 mW/cm ²	MA	70 73	1.25 1.3	0.2 mm A4 paper 5.0 mm pig skin	8
RTPP	PET- RAFT	740 nm, 66 mW/cm ²	MMA	50	15	7 mm pig skin	20

Reference

57. Wu, C. Jung, K. Ma, Y. Liu, W. & Boyer, C. Unravelling an oxygen-mediated reductive quenching pathway for photopolymerisation under long wavelengths. *Nat. Commun.* **12**, 478 (2021).

Supplementary Reference

5. Jiang, J. et al. Localized surface plasmon resonance meets controlled/living radical polymerization: an adaptable strategy for broadband light-regulated macromolecular synthesis. *Angew. Chem. Int. Ed.* **58**, 12096-12101 (2019).

6. Allison-Logan, S. et al. From UV to NIR: a full-spectrum metal-free photocatalyst for efficient polymer synthesis in aqueous conditions. *Angew. Chem. Int. Ed.* **59**, 21392-21396 (2020).

8. Wu, Z. Jung, K. & Boyer, C. Effective utilization of NIR wavelengths for photo-controlled polymerization: penetration through thick barriers and parallel solar syntheses. *Angew. Chem. Int. Ed.* **59**, 2013-2017 (2020).

14. Kütahya, C. Schmitz, C. Strehmel, V. Yagci, Y. & Strehmel, B. Near-infrared sensitized photoinduced atom-transfer radical polymerization (ATRP) with a copper (II) catalyst concentration in the ppm range. *Angew. Chem. Int. Ed.* **57**, 7898-7902 (2018).
18. Shanmugam, S. Xu, J. & Boyer, C. Light-regulated polymerization under near-infrared/far-red irradiation catalyzed by bacteriochlorophyll *a*. *Angew. Chem. Int. Ed.* **128**, 1048-1052 (2016).
19. Wu, C. Jung, K. Ma, Y. Liu, W. & Boyer, C. Unravelling an oxygen-mediated reductive quenching pathway for photopolymerisation under long wavelengths. *Nat. Commun.* **12**, 478(2021).
22. Cao, H. et al. Far-red light-induced reversible addition-fragmentation chain transfer polymerization using a man-made bacteriochlorin. *ACS Macro Lett.* **8**, 616-622 (2019).
21. Qiao, X. et al. Simple full-spectrum heterogeneous photocatalyst for photo-induced atom transfer radical polymerization (ATRP) under UV/vis/NIR and its application for the preparation of dual mode curing injectable photoluminescence hydrogel. *ACS Appl. Mater. Interfaces* **14**, 21555-21563 (2022).
- Corrigan, N. Xu, J. T. & Boyer, C. A Photoinitiation system for conventional and controlled radical polymerization at visible and NIR wavelengths. *Macromolecules* **49**, 3274-3285 (2016).
23. Zhang, W. et al. Atom transfer radical polymerization driven by near-infrared light with recyclable upconversion nanoparticles. *Macromolecules* **53**, 4678-4684 (2020).
24. Zhu, Y. Liu, Y. Miller, K. A. Zhu, H. & Egap, E. Lead halide perovskite nanocrystals as photocatalysts for PET-RAFT polymerization under visible and near-infrared irradiation. *ACS Macro Lett.* **9**, 725-730 (2020).
25. Xia, Z. N. Shi, B. F. Zhu, W. J. Xiao, Y. & Lu, C. L. Binary hybridization strategy toward stable porphyrinic Zr-MOF encapsulated perovskites as high-performance heterogeneous photocatalysts for red to NIR light-induced PET-RAFT polymerization. *Adv. Funct. Mater.* **32**, 2207655 (2022).
26. Ding, C. et al. Platform of near-infrared light-induced reversible deactivation radical polymerization: upconversion nanoparticles as internal light sources. *Polym. Chem.* **7**, 7370-7374 (2016).
27. Hu, L. J. et al. The *in situ* “grafting from” approach for the synthesis of polymer brushes on upconversion nanoparticles via NIR-mediated RAFT polymerization. *Polym. Chem.* **12**, 545 (2021).
28. Zhang, J. et al. From 0-dimension to 1-dimensions: Au nanocrystals as versatile plasmonic photocatalyst for broadband light induced RAFT polymerization. *Polym. Chem.* **12**, 2439-2446

(2021).

29. Li, M. et al. Dual enhancement of carrier generation and migration on Au/g-C₃N₄ photocatalysts for highly-efficient broadband PET-RAFT polymerization. *Polym. Chem.* **13**, 1022-1030 (2022).

6) Some of the characterization of the hyperbranched polymers in SI could be moved in main text.

Response: Thank you very much for your valuable suggestion. We have moved the characterizations (**Supplementary Figure 3, 5, 10, 13**) to the main text.

The manuscript was revised as follows:

Figure Captions

Fig. 2 | The structure characterization and photoelectric performance of the PPh₃-CHCP. **a**, Solid-state ¹³C NMR spectrum of PPh₃-HCP. Asterisks denote spinning sidebands. **b**, Solid-state ³¹P NMR spectrum of PPh₃-HCP and PPh₃-CHCP respectively. **c**, and **d**, Relative XPS survey spectra of PPh₃-HCP and PPh₃-CHCP respectively. **e**, UV-Vis-NIR diffuse reflectance spectrum of the photocatalyst overlaid with the emission spectra of the light sources, including blue ($\lambda_{\text{max}} = 455$ nm), green ($\lambda_{\text{max}} = 540$ nm), orange ($\lambda_{\text{max}} = 590$ nm), red ($\lambda_{\text{max}} = 630$ nm), 730 nm, 760 nm, 800 nm, 850 nm, 940 nm, and white light respectively. **f**, Photocurrent response curve of PPh₃-CHCP.

Reviewer 3

This manuscript describes the development of crosslinked polymers (PPh₃-CHCP) from PPh₃ and p-dimethoxybenzene, which can be used as an effective heterogeneous photocatalyst to activate the Cu-catalyzed ATRP. This photo and copper co-catalytic system showed a high efficiency under the irradiation of a wide range of light (450-940 nm). Polymerization at higher volumes (200-400 mL) was also successfully demonstrated. The high efficiency under NIR light is quite impressive. Overall, this work could be suitable for publication after addressing the following concerns:

Response: We appreciate the positive comments and suggestions that helped us to significantly improve the manuscript.

1. The known development of heterogeneous photocatalysts via conjugated cross-linked polymers for activating Cu-catalyzed ATRP (e.g. Ref. 41) as well as examples of ATRP under red light (e.g. ACS Macro Lett. 2022, 11, 3, 376) should be mentioned and discussed.

Response: Thank you very much for your valuable suggestion. Benefitting from heterogeneous nature, photocatalysts such as conjugated microporous polymers of phenothiazine (PTZ-CMP) can not only enable easy separation and efficient reusability in Cu-ATRP, but also produce polymers with high conversion and well-controlled molecular weight under green light irradiation (*J. Am. Chem. Soc.* **143**, 9630-9638 (2021)). Photo-ATRP using Zinc(II) Tetraphenylporphine (ZnPor) as photocatalyst offered well-controlled polymerizations without the need for deoxygenation processes under red light irradiation (*ACS Macro Lett.* **11**, 376 (2022)). More importantly, the development of dual photoredox catalytic systems endowed Cu-ATRPs with fascinating properties such as easy catalyst separation, oxygen tolerance, and long wavelength light activation, etc. (*J. Am. Chem. Soc.* **143**, 9630-9638 (2021); *ACS Macro Lett.* **11**, 376 (2022)) Generally, photocatalyzed RDRP under long wavelength light could suppress side reactions and promote photon penetration in reaction media (*Angew. Chem. Int. Ed.* **58**, 12096-12101 (2019)). Therefore, developing suitable photocatalysts

for dual photoredox catalytic systems could greatly help to achieve NIR light-induced Cu-ATRP (*Angew. Chem. Int. Ed.* **57**, 7898-7902 (2018)).

The manuscript was revised as follows:

Introduction

Benefitting from heterogeneous nature, photocatalysts such as conjugated microporous polymers of phenothiazine (PTZ-CMP) can not only enable easy separation and efficient reusability in Cu-ATRP, but also offer polymers with high conversion and well-controlled molecular weight under green light irradiation⁴⁴.

Results

Performing photocatalyzed RDRP under long wavelength light can suppress side reactions and enhance photon penetration depth in reaction media⁴⁹. The development of dual photoredox catalytic systems with Cu-ATRP such as the use of zinc(II) tetraphenylporphine (ZnPor) photocatalyst offer opportunities to produce polymers with high yielding under long wavelength light irradiation^{44,57}. Therefore, developing suitable photocatalysts for dual photoredox catalytic systems could greatly help to achieve NIR light-induced Cu-ATRP⁵⁵.

Reference

57. Dadashi-Silab, S. et al. Red-light-induced, copper-catalyzed atom transfer radical polymerization. *ACS Macro Lett.* **11**, 376-381 (2022).

2. ‘... has remained a big challenge. There was also no report on large scale photo-induced production of block copolymers’ The tone of this sentence should be attenuated a little bit, as some known catalytic systems (e.g. PET-RAFT polymerizations) as well as flow chemistry can be applied to a larger scale synthesis.

Response: We agree with your suggestion. As reported, several photo RDRP systems have shown marvelous prospects of scaling up reactions with the rapid development of microfluidic engineering (*Angew. Chem. Int. Ed.* **57**, 14260-14264 (2018); *Polym. Chem.* **10**, 4402-4406 (2019); *React. Chem. Eng.* **4**, 1216-1228 (2019);

Macromolecules **52**, 5611-5617 (2019); *Nat. Commun.* **13**, 3918 (2022); *Sci. Chin. Chem.* **64**, 844-851 (2021); *Angew. Chem.* **134**, e202116135 (2022)). Related products such as PAO-*b*-PPEGMA showed great potential application in boosting uranium harvesting from seawater (*Nat. Commun.* **13**, 3918 (2022)).

The manuscript was revised as follows:

Meanwhile, the efforts development of materials/methodology for large scale synthesis with a good control over the polymer's molecular weight and dispersity have always been on rise. As reported, several photo RDRP systems have shown marvelous prospects of scaling up reactions with the rapid development of microfluidic engineering²⁸⁻³⁰. Related products such as PAO-*b*-PPEGMA showed great potential application in boosting uranium harvesting from seawater²⁹. From the industrial application point of view, large-scale production of block copolymers is clearly an area worth continuous exploring.

Reference

28. Lauterbach, F. Rubens, M. Abetz, V. & Junkers T. Ultrafast photoRAFT block copolymerization of isoprene and styrene facilitated through continuous-flow operation. *Angew. Chem. Int. Ed.* **57**, 14260-14264 (2018).

29. Liu, Z. et al. Multi-scale computer-aided design and photo-controlled macromolecular synthesis boosting uranium harvesting from seawater. *Nat. Commun.* **13**, 3918 (2022).

30. Chen, K. Zhou, Y. Han, S. Liu, Y. & Chen, M. Main-chain fluoropolymers with alternating sequence control via light-driven reversible-deactivation copolymerization in batch and flow. *Angew. Chem.* **134**, e202116135 (2022).

3. In Fig. 1 b, the TOC graphic, as well as several places in the main text, the copper (Cu) catalysts should be mentioned with PPh₃-CHCP together, otherwise, could be a little bit misleading and give the readers a feeling that the ATRP was catalyzed by PPh₃-CHCP alone without Cu/Me₆TREN.

Response: Thank you very much for your valuable comment! As suggested, we revised the main text, figures, and TOC as follows:

Main text

Abstract

Herein, we report the development of a phosphine-based conjugated hypercrosslinked polymer (PPh₃-CHCP) as a photocatalyst for an efficient photoinduced copper-catalyzed atom transfer radical polymerization (Cu-ATRP) with O₂ tolerance.

The sunlight-driven Cu-ATRP allowed synthesis of homopolymers at 200 mL from various monomers, and the monomer conversion approached 99% in cloud intermittency with good control over polydispersity respectively.

Introduction

In this regard, we report the synthesis of phosphine-based conjugated hyper crosslinked polymer (PPh₃-CHCP) photocatalyst and its application for the first persistent large-scale sunlight-driven Cu-ATRP with O₂ tolerance (Fig. 1).

Conclusion

Sunlight-driven Cu-ATRPs were performed from MA and MMA respectively.

Fig. 1 | Development of photocatalyst for large scale sunlight-driven Cu-ATRP. c, Photoinduced Cu-ATRP in the presence of PPh₃-CHCP (inserted photo: 200mL reaction scale of PMA under sunlight irradiation).

TOC:

4. Supplementary Table 6, without $\text{PPh}_3\text{-CHCP}$, 48% conversion of MA was also observed. The authors should comment on this result and perform more control experiments with MA under blue and green light.

Response: Thank you very much for your valuable comment. In control experiments, polymerizations of MA were performed under blue and green light irradiation without $\text{PPh}_3\text{-CHCP}$ to evaluate potential background reaction. As shown in Supplementary Table 11, negligible conversion of MA was observed under blue and green light irradiation (according to ^1H NMR and GPC analysis), suggesting the concentration of Cu(I) activator regenerated by Cu(II) was insufficient in predetermined time (3 h for blue light and 8 h for green light). In our sunlight-induced polymerizations, the UV may reduce Cu(II) species and subsequently lead to the conversion of MA. This is in consistent with the previous reports (e.g., *J. Am. Chem. Soc.* **136**, 13303-13312 (2014)). The manuscript was revised as follows:

Considering there was negligible conversion of MA under blue and green light irradiation (Supplementary Table 11), the concentration of Cu^{I} activator regenerated by UV rays from sunlight was not sufficient to sustain significant chain growth in a short period⁵⁹.

Supplementary Table 11. Results of control experiments in polymerization of MA without PPh₃-CHCP under blue and green light irradiation respectively.

entry	Light	Time (h)	Conv. (%)	$M_{n,th}$	M_n	\bar{D}
1	Blue LED	3	0	-	-	-
2	Green LED	8	0	-	-	-

Reaction conditions: [MA]/[EBiB]/[CuBr₂]/[Me₆TREN] = 200/1/0.04/0.12 in 50 vol% DMSO under blue or green light irradiation (0.9 mW/cm²) for preset time interval at ambient temperature.

Reference

59. Ribelli, T. G. Konkolewicz, D. Bernhard, S. & Matyjaszewski, K. How are radicals (re) generated in photochemical ATRP? *J. Am. Chem. Soc.* **136**, 13303-13312 (2014).

5. In Fig. 2 C, the results showed no apparent conversion in the dark periods. How about a prolonged dark period (5 h or 12 h) after initiation. The best is also to provide a plot of conversion over time in the Supplementary Information.

Response: This is a very important suggestion, As suggested, we conducted polymerization of MA with a dark period as long as 12 h after initiation. Negligible monomer conversion was observed in dark period (Supplementary Figure 20), and this suggested that active radicals could not be efficiently generated under dark conditions. The manuscript was revised as follows:

The dark period was prolonged up to 12 h and related results are shown in Supplementary Fig. 20. A negligible monomer conversion was observed in dark, which suggest that active radicals are not efficiently generated under dark conditions.

Supplementary Figure 20. Plot of monomer conversion versus time on the exposure time under green light irradiation (0.9 mW/cm^2) that was switched on and off in the presence of the $\text{PPh}_3\text{-CHCP}$ photocatalyst. Reaction conditions: $[\text{MA}]/[\text{EBiB}]/[\text{CuBr}_2]/[\text{Me}_6\text{TREN}] = 200/1/0.04/0.2$ in 50 vol % DMF, $\text{PPh}_3\text{-CHCP} = 2 \text{ mg/mL}$.

6. The light sources (emission spectra) as well as details about the ATRP experiment under NIR should be provided in the Supplementary Information.

Response: Thank you very much for your valuable suggestions. Accordingly, the light sources (emission spectra) as well as other conditions about the ATRPs under NIR are provided.

The Supplementary Information was revised as follows:

General procedure for NIR photoinduced ATRP of methyl acrylate

NIR photoinduced ATRP process of methyl acrylate using $\text{PPh}_3\text{-CHCP}$ were conducted with varied amount of photocatalyst, monomer, solvent (DMSO), catalyst ($\text{CuBr}_2/\text{Me}_6\text{TREN}$), and initiator (EBiB). Typical synthesis of PMA was as follows: photocatalyst $\text{PPh}_3\text{-CHCP}$ (1.6 mg), MA (810 μL , 9.0 mmol, 200 equiv.), DMSO (810 μL), and a stock solution of CuBr_2 (0.4 mg, 1.8 μmol , 0.04 equiv.), and Me_6TREN (2.4 μL , 9.0 μmol , 0.2 equiv.) in DMSO (20 μL) were added to a Schlenk tube under nitrogen atmosphere. The tube equipped with a magnet bar was sealed with a rubber septum and degassed by three freeze-vacuum-thaw cycles. A 6.6 μL aliquot of EBiB (45.0 μmol , 1 equiv.) was introduced into the tube via syringe. The tube was irradiated under 940 nm (15 mW/cm^2) to start the polymerization. Samples were taken

periodically and analyzed by ^1H NMR and SEC to determine the monomer conversion and molecular weight properties, respectively. PMA can be obtained after filtration of $\text{PPh}_3\text{-CHCP}$ followed by precipitation in methanol directly.

General procedure NIR photoinduced ATRP penetration experiment using A4 paper as barrier

Typical procedure for NIR photoinduced ATRP penetration experiment using A4 paper as barrier was as follows: the photocatalyst $\text{PPh}_3\text{-CHCP}$ (3.2 mg), MA (810 μL , 9.0 mmol, 200 equiv.), DMSO (810 μL), and a stock solution of CuBr_2 (0.2 mg, 0.9 μmol , 0.02 equiv.), and Me_6TREN (2.4 μL , 9.0 μmol , 0.2 equiv.) in DMSO (20 μL) were added to a Schlenk tube under nitrogen atmosphere. The tube equipped with a magnet bar was sealed with a rubber septum and degassed by three freeze-vacuum-thaw cycles. A 6.6 μL aliquot of EBiB (45.0 μmol , 1 equiv.) was introduced into the tube via syringe. The tube was exposed to 940 nm LEDs (30 mW/cm^2) with 0.1 mm paper (1 A4 paper) as the barrier for 6 h. ^1H NMR and SEC to determine the monomer conversion and molecular weight properties respectively. PMA can be obtained after filtration of $\text{PPh}_3\text{-CHCP}$ followed by precipitation in methanol directly.

Fig. 2 | The structure characterization and photoelectric performance of the $\text{PPh}_3\text{-CHCP}$. (e) UV-Vis-NIR diffuse reflectance spectrum of the photocatalyst overlaid with the emission spectra of the light sources, including blue ($\lambda_{\text{max}} = 455$ nm), green ($\lambda_{\text{max}} = 540$ nm), orange ($\lambda_{\text{max}} = 590$ nm), red ($\lambda_{\text{max}} = 630$ nm), 730 nm, 760 nm, 800 nm, 850 nm, 940 nm, and white light respectively.

Reviewers' Comments:

Reviewer #1:

Remarks to the Author:

The authors have done an excellent job at addressing all of the reviewer's comments thoroughly. This manuscript can now be accepted in the current form.

Reviewer #2:

Remarks to the Author:

the authors have answered the reviewers' comments. I recommend publication as is.

Reviewer #3:

Remarks to the Author:

In this revision, substantial improvement has been made, but I still have some concerns:

- The title could be a little misleading, and the abstract also has the same problem. The title "Conjugated Cross-linked Phenothiazines as Green or Red Light Heterogeneous Photocatalysts for Copper-Catalyzed Atom Transfer Radical Polymerization" by Matyjaszewski et al. (J. Am. Chem. Soc. 2021, 143, 25, 9630–9638) is a good example, which could clearly show the role of polymer photocatalysts and the nature of the ATRP.

-In Table 1, I saw the footnote "x = 0.04, 0.02, 0.01, and 0.005 corresponding to 200, 100, 50, and 25 ppm with respect to monomer; y = 0.04, 0.08, 0.12, 0.16 or 0.2)", but I could not find the corresponding results in the table. Did the authors run the reaction at 25 ppm of catalyst loading? In the discussion, only 200 ppm was mentioned.

- Several places like "the illumination intensity of sunlight is generally higher than 1000 W m⁻²", "under sunlight is far less efficient" are also a little bit misleading. The authors should avoid the use of "generally" "far", which could make the meanings deviate to some extent from the truth.

- Regarding O₂ tolerance, the authors may need to turn down the tone further. "A Robust and Versatile Photoinduced Living Polymerization of Conjugated and Unconjugated Monomers and Its Oxygen Tolerance" by Boyer et al. (J. Am. Chem. Soc. 2014, 136, 14, 5508–5519) is a good example. The current results only indicated the system could tolerate some amount of O₂ rather than an open-air experiment. In Boyer's paper, they also mentioned the beneficial effect of using DMSO to improve O₂ tolerance.

- Fig 3. c is not the Plot of monomer conversion versus time. The author may include the plot of monomer conversion as well as the operation details of this experiment (e.g. the time taking the samples) in the SI.

- In the SI, the author proposed P=O included in the PPh₃-CHCP; is there any support to the formation of P=O and the mechanism? If so, a short description of this structure information in the main text could be nice for the readers.

Given below are our point-by-point responses (in **BLUE** colour) to the reviewers' comments. The changes to the manuscript and supplementary information are marked in **RED** colour.

Reviewer 1

The authors have done an excellent job at addressing all of the reviewer's comments thoroughly. This manuscript can now be accepted in the current form.

Response: We appreciate the reviewer's comments and suggestions that helped us to significantly improve the manuscript.

Reviewer 2

The authors have answered the reviewers' comments. I recommend publication as is.

Response: We appreciate the reviewer's comments and suggestions that helped us to significantly improve the manuscript.

Reviewer 3

In this revision, substantial improvement has been made, but I still have some concerns:

Response: We appreciate the positive comments and suggestions that helped us to significantly improve the manuscript.

1. The title could be a little misleading, and the abstract also has the same problem. The title "Conjugated Cross-linked Phenothiazines as Green or Red Light Heterogeneous Photocatalysts for Copper-Catalyzed Atom Transfer Radical Polymerization" by Matyjaszewski et al. (J. Am. Chem. Soc. 2021, 143, 25, 9630-9638) is a good example, which could clearly show the role of polymer photocatalysts and the nature of the ATRP.

Response: Thank you very much for your valuable suggestions. As suggested, we have revised the title as "Conjugated Cross-linked Phosphine as Broadband Light or Sunlight-Driven Photocatalyst for Large-Scale Atom Transfer Radical Polymerization".

Title

Conjugated Cross-linked Phosphine as Broadband Light or Sunlight-Driven Photocatalyst for Large-Scale Atom Transfer Radical Polymerization

We revised the abstract as follows:

Abstract

The use of light to regulate photocatalyzed reversible deactivation radical polymerization (RDRP) under mild conditions, especially driven by broadband light or sunlight directly, is highly desired. But the development of a suitable photocatalyzed polymerization system for large-scale production of polymers, especially block copolymers, has remained a big challenge. Herein, we report the development of a phosphine-based conjugated hypercrosslinked polymer (PPh₃-CHCP) photocatalyst for an efficient large-scale photoinduced copper-catalyzed atom transfer radical polymerization (Cu-ATRP). Monomers including acrylates and methyl acrylates can achieve near-quantitative conversions under a wide range (450-940 nm) of radiations or sunlight directly. The photocatalyst could be easily recycled and reused. The sunlight-driven Cu-ATRP allowed the synthesis of homopolymers at 200 mL from various monomers, and monomer conversions approached 99% in clouds intermittency with good control over polydispersity. In addition, block copolymers at 400 mL scale can also be obtained, which demonstrates its great potential for industrial applications.

2. ‘In Table 1, I saw the footnote “x = 0.04, 0.02, 0.01, and 0.005 corresponding to 200, 100, 50, and 25 ppm with respect to monomer; y = 0.04, 0.08, 0.12, 0.16 or 0.2)”, but I could not find the corresponding results in the table. Did the authors run the reaction at 25 ppm of catalyst loading? In the discussion, only 200 ppm was mentioned.

Response: Thank you very much for your valuable suggestions. We have now revised the footnote to better understand the Table 1. In entries 18 and 21, experiments were performed with 25 ppm of copper catalyst loading. We are sorry for missing the discussion on the relationship between copper concentration and polymerization. Decreasing the concentration of CuBr₂ resulted in an increase in the rate of polymerization but produced polymers with relatively broad dispersity values. For

example, fast polymerization of MA was achieved that offered 99% monomer conversion within <5 h but with a high dispersity value of 1.20 (entry 18, Table 1), by reducing the amount of CuBr₂ from 0.04 to 0.005 equiv (with respect to initiator or 25 ppm with respect to monomer) (Supplementary Fig. 16). In the presence of 0.01 or 0.02 equiv (50 or 100 ppm, respectively) of CuBr₂ (entries 19 and 20, Table 1), the resultant polymers showed a low dispersity of 1.10 (Supplementary Fig. 16).

Accordingly, we revised Table 1 in the manuscript as follows:

entry	ligand	CuBr ₂ /L	PPh ₃ -CHCP (mg/mL)	solvent	time (h)	conv (%)	$M_{n,th}$	M_n	\mathcal{D}
1	Me ₆ TREN	1/1	0.5	DMSO	8	0	-	-	-
2	Me ₆ TREN	1/2	0.5	DMSO	8	0	-	-	-
3	Me ₆ TREN	1/3	0.5	DMSO	8	99	17300	17600	1.06
4	Me ₆ TREN	1/4	0.5	DMSO	8	99	17300	18700	1.06
5	Me ₆ TREN	1/3	0.25	DMSO	10	99	17300	17500	1.05
6	Me ₆ TREN	1/3	0.125	DMSO	12	99	17300	18200	1.07
7	Me ₆ TREN ^a	1/3	0.5	DMSO	12	99	43300	43600	1.09
8	Me ₆ TREN	1/3	0.5-in dark	DMSO	12	0	-	-	-
9	Me ₆ TREN	1/3	0	DMSO	12	0	-	-	-
10	Me ₆ TREN	1/5	2	MeCN	24	94	16400	15700	1.06
11	Me ₆ TREN	1/5	0	MeCN	24	0	-	-	-
12	Me ₆ TREN	1/5	2	DMF	20	99	17300	17000	1.05
13	Me ₆ TREN	1/5	0	DMF	20	0	-	-	-
14	TPMA ^b	1/5	0.5	DMSO	40	93	16200	15600	1.07
15	TPMA ^b	1/5	0-in dark	DMSO	24	0	-	-	-
16	PMDETA	1/5	1	DMSO	36	90	15700	15500	1.12
17	PMDETA	1/5	0	DMSO	36	0	-	-	-
18	Me ₆ TREN	0.125/3	0.25	DMSO	5	99	17300	17200	1.20
19	Me ₆ TREN	0.25/3	0.25	DMSO	5	99	17300	17100	1.10
20	Me ₆ TREN	0.5/3	0.25	DMSO	5	99	17300	17900	1.10
21	Me ₆ TREN	0.125/3	0.125	DMSO	12	99	17300	18200	1.07

Polymerizations were conducted in different solvents (50 vol %) and irradiated under green light (0.9 mW/cm²). In entry 1-17, [MA]/[EBiB]/[CuBr₂]/[L] = 200/1/0.04/x (L = Me₆TREN, TPMA, or PMDETA; x = 0.04, 0.08, 0.12, 0.16, or 0.2). In entry 18-21, [MA]/[EBiB]/[CuBr₂]/[Me₆TREN] = 200/1/y/0.12 (y = 0.005, 0.01, and 0.02 corresponding to 25, 50, and 100 ppm with respect to the monomer). ^aDP=500. ^bTriethanolamine (0.6 equiv relative to EBiB) was used as the electron donor in the presence of TPMA. *Abbreviations: MA* methyl acrylate, *Conv.* conversion, $\mathcal{D} = M_w/M_n$, *Me₆TREN* tris[2-(dimethylamino)ethyl]amine, *TPMA* tris(2-pyridylmethyl)amine, *PMDETA* N,N,N',N'',N'''-pentamethyldiethylenetriamine, *DMSO* dimethylsulfoxide,

DMF N,N-dimethylformamide, *MeCN* acetonitrile, *EBiB* ethyl α -bromoisobutyrate, *DP* degree of polymerization.

We revised the manuscript as follows:

Main Text

Decreasing concentration of CuBr₂ resulted in an increase in the rate of polymerization, but produced polymers with relatively broad dispersity. For example, fast polymerization of MA was achieved that offered 99% monomer conversion within <5 h but with a high dispersity value of 1.20 (entry 18, Table 1), by reducing the amount of CuBr₂ from 0.04 to 0.005 equiv (with respect to initiator or 25 ppm with respect to monomer) (Supplementary Fig. 16). In the presence of 0.01 or 0.02 equiv (50 or 100 ppm, respectively) of CuBr₂ (entries 19 and 20, Table 1), the resultant polymers showed a low dispersity of 1.10 (Supplementary Fig. 16).

3. Several places like “the illumination intensity of sunlight is generally higher 1000 W m⁻²”, “under sunlight is far less efficient” are also a little bit misleading. The authors should avoid the use of “generally” “far”, which could make the meanings deviate to some extent from the truth.

Response: This is a good suggestion! We have checked and now removed the use of “generally” and “far” in the revised manuscript.

We revised the manuscript as follows:

Introduction

For example, in practical applications, different from laboratory light sources, the illumination intensity of sunlight is much higher³¹.

In organocatalyzed ATRP, polymerization performed under sunlight is less efficient than that under white light^{39,40}.

4. Regarding O₂ tolerance, the authors may need to turn down the tone further. “A Robust and Versatile Photoinduced Living Polymerization of Conjugated and Unconjugated Monomers and Its Oxygen Tolerance” by Boyer et al. (J. Am. Chem. Soc. 2014, 136, 14, 5508-5519) is a good example. The current results only indicated

the system could tolerate some amount of O₂ rather than an open-air experiment. In Boyer's paper, they also mentioned the beneficial effect of using DMSO to improve O₂ tolerance.

Response: Thanks much for pointing it out! Previous studies have revealed that DMSO could also react with singlet oxygen derived from the transformation of oxygen to generate sulfone (*Macromolecules* **49**, 6779-6789 (2016); *Angew. Chem. Int. Ed.* **58**, 12096-12101 (2019)). Hence, following the previous report (*Acc. Chem. Res.* **54**, 1779-1790 (2021)), we revised the related description.

Introduction

In this regard, we report the synthesis of phosphine-based conjugated hyper crosslinked polymer (PPh₃-CHCP) photocatalyst and its application for the first persistent large-scale sunlight-driven Cu-ATRP with limited O₂ tolerance (without deoxygenation procedure) (Fig. 1).

Main Text

PPh₃-CHCP catalyzed polymerization also demonstrated limited oxygen tolerance.

Discussion

Near-quantitative monomer conversions could be obtained under a single sunlight irradiation period (6 h), and these polymerizations exhibited limited tolerance to O₂ (without deoxygenation procedure).

5. Fig 3. c is not the Plot of monomer conversion versus time. The author may include the plot of monomer conversion as well as the operation details of this experiment (e.g. the time taking the samples) in the SI.

Response: Thank you very much for pointing this out. We addressed this mistake in Figure 3, and the experimental details were also provided in the revised manuscript.

Fig. 3 | Results for the polymerization of MA using PPh₃-CHCP. (c) Temporal control in ATRP of MA upon intermittent switching green light on/off in the presence of the PPh₃-CHCP photocatalyst.

General procedure for temporal control in photoinduced ATRP of MA using PPh₃-CHCP under green light irradiation

The photocatalyst PPh₃-CHCP (1.6 mg), MA (1.62 mL, 18.0 mmol, 200 equiv.), DMF (1.62 mL), a stock solution of CuBr₂ (0.8 mg, 3.6 μmol, 0.04 equiv.), and Me₆TREN (4.8 μL, 18.0 μmol, 0.2 equiv.) in DMF (40 μL) were added to a Schlenk tube under nitrogen atmosphere. The tube equipped with a magnet bar was sealed with a rubber septum and degassed by three freeze-vacuum-thaw cycles. A 13.2 μL aliquot of EBiB (90.0 μmol, 1 equiv.) was introduced into the tube *via* syringe. The tube was irradiated under green LEDs to start the polymerization. After 6 h exposure, the reaction tube was kept in dark for 2 h and exposed to repeated cycles for 2 h. In these subsequent intervals, 0.2 mL of reaction mixture were syringed out from the polymerization media and precipitated in methanol. Samples were analyzed by ¹H NMR and SEC to determine the monomer conversion and molecular weight properties, respectively.

6. In the SI, the author proposed P=O included in the PPh₃-CHCP; is there any support to the formation of P=O and the mechanism? If so, a short description of this instruction information in the main text could be nice for the readers.

Response: Thank you very much for your valuable suggestion. The binding energy of P in PPh₃-CHCP was at around 133.1 eV according to XPS analysis, which was in

agreement with that of P in triphenylphosphine oxide complex according to previous literatures (*Adv. Mater.* **28**, 4824-4831 (2016); *Adv. Mater.* **31**, 1805944 (2019); *ACS Energy Lett.* **6**, 4265-4272 (2021)). As reported, ferric chloride may react with triphenylphosphine in acetonitrile or other solvents to offer $\text{FeCl}_3(\text{OPPh}_3)$, and a complete oxidation of triphenylphosphine in the solid $\text{FeCl}_3(\text{PPh}_3)_2$ would take place at 80°C in the presence of air oxygen (*Chem. zvesti.* **30**, 86-89 (1976); *Org. Lett.* **20**, 7419–7423 (2018)). The dissolved oxygen in the reaction solutions may be helpful in the formation of $\text{P}=\text{O}$.

Accordingly, we have revised the manuscript as follows:

The binding energy of P in $\text{PPh}_3\text{-CHCP}$ was found to be 133.1 eV, indicating the formation of $\text{P}=\text{O}$, which may be resulted from the oxidation reaction between ferric chloride and triphenylphosphine in the presence of dissolved oxygen in the reaction media⁵¹.

Reference

51. Aubineau, T. & Cossy, J. A one-pot reaction toward the diastereoselective synthesis of substituted morpholines. *Org. Lett.* **20**, 7419-7423 (2018).